# Solving Parameter-Robust Avoid Problems with Unknown Feasibility using Reinforcement Learning

**Oswin So**[*1]  **Eric Yang Yu**[*1]  **Songyuan Zhang**[1]
**Matthew Cleaveland**[2]  **Mitchell Black**[2]  **Chuchu Fan**[1]
[1]Department of Aeronautics and Astronautics, MIT    [2]MIT Lincoln Laboratory
{oswinso,eyyu}@mit.edu

## Abstract

Recent advances in deep reinforcement learning (RL) have achieved strong results on high-dimensional control tasks, but applying RL to optimal safe controller synthesis raises a fundamental mismatch: optimal safe controller synthesis seeks to maximize the set of states from which a system remains safe indefinitely, while RL optimizes expected returns over a user-specified distribution. This mismatch can yield policies that perform poorly on low-probability states still within the safe set. A natural alternative is to frame the problem as a robust optimization over a set of initial conditions that specify the initial state, dynamics and safe set, but whether this problem has a solution depends on the feasibility of the specified set, which is unknown a priori. We propose **Feasibility-Guided Exploration (FGE)**, a method that simultaneously identifies a subset of feasible initial conditions under which a safe policy exists, and learns a policy to solve the optimal control problem over this set of initial conditions. Empirical results demonstrate that FGE learns policies with over $50\%$ more coverage than the best existing method for challenging initial conditions across tasks in the MuJoCo simulator. The project page can be found at https://oswinso.xyz/fge. [1]

## 1 Introduction

Hamilton-Jacobi (HJ)-based optimal control analysis is a powerful tool for obtaining control policies and reasoning about the largest set of initial conditions from which a dynamical system can satisfy safety constraints for all time (Lygeros et al., 1998). However, classical methods for solving HJ-based optimal control analysis use grid-based solvers that require state discretization and hence suffer from the curse of dimensionality, rendering them intractable for systems with high-dimensional nonlinear dynamics (Mitchell et al., 2005). Recent advances in deep reinforcement learning (RL) have demonstrated remarkable success in approximating optimal control solutions in high dimensions (Fisac et al., 2019; Hsu et al., 2021; Bansal & Tomlin, 2021; Ganai et al., 2023; Hsu et al., 2023; Nguyen et al., 2024; So et al., 2024; Ganai et al., 2024). In these approaches, one typically poses an optimization problem whose exact solution is the optimal control solution for a single initial condition, then uses RL to learn a policy that solves this problem over some user-defined distribution of initial conditions. However, this sampling-based formulation introduces a fundamental misalignment: the RL objective minimizes expected cost under a potentially narrow initial distribution rather than worst-case safety over the entire set of interest. As a result, learned policies may perform well on frequently sampled states according to the user-specified distribution, but fail catastrophically in low-density regions that are still within the operating region and could be made safe.

A natural remedy is to recast the HJ-based optimal control problem as a robust optimization problem: Given a set of initial conditions of interest in the form of a set of initial states (and more generally, a set of *parameters* that specify the initial state, dynamics and safety specification), instead of arbitrarily introducing a distribution over this set and taking an expectation, one minimizes the cost under the worst-case initial condition within this set (Pinto et al., 2017). This perspective aligns directly with the original goal of worst-case safety. However, it hinges on being able to specify a candidate set of initial conditions that are feasible (i.e., can be made safe), which in itself is a nontrivial task, as finding

---

*These authors contributed equally to this work.
[1]DISTRIBUTION STATEMENT A. Approved for public release. Distribution is unlimited.

Figure 1: **FGE modifies the parameter sampling distribution.** FGE modifies the **base** distribution over the parameters by introducing an **explore** distribution to improve policy performance on parameters that have not been observed to be safe so far, and a **rehearsal** distribution obtained via sampling-based approximate **best response** to train on poorly performing parameters that were previously solved. We combine all three distributions to obtain the final distribution for **FGE** that balances the objectives of maximizing safety rate gain and minimizing safety rate loss.

the set of feasible initial conditions is precisely the goal of HJ-based optimal control analysis and is computationally infeasible to solve exactly (Mitchell et al., 2007). An infeasible robust problem renders all policies equally optimal (as all yield the same worst-case objective), which does not solve the original safety problem.

In this paper, we address this gap by introducing the *parameter-robust avoid problem with unknown feasibility*. Our key idea is to simultaneously (i) identify a large subset of parameters where safety is achievable, and (ii) learn a policy satisfying the safety constraints on this identified subset (Fig. 1).

Our main contributions are as follows:

- We formalize the novel robust safe optimal control objective that jointly selects a maximal feasible parameter region and a safe policy.
- We propose a practical framework that interleaves (a) feasibility-guided set expansion via constraint-driven exploration and (b) robust policy optimization over the current feasible set using techniques from saddle-point finding and online learning.
- On a suite of high-dimensional avoid problems in the MuJoCo simulator, our approach learns policies that yield significantly larger safe sets than prior methods.

## 2 RELATED WORK

**Saddle-point Problems** Solving a robust safe optimal control problem requires the solution of a minimax optimization problem, which results in a saddle-point problem if the equivalent game has a Nash equilibrium (Orabona, 2019). Recent works have shown connections between saddle-point problems and online learning when players play with sublinear regret (Orabona, 2019). These methods have been used in RL to solve for game theoretic equilibria in large-scale games such as poker (Brown et al., 2019) and Diplomacy (Gray et al., 2020). The parameter-robust avoid problem with unknown feasibility differs from existing works in saddle-point finding in that the domain for one of the players is an optimization variable and is constrained to be a subset of the feasible space that is not known a priori and must be solved for as well.

**Robust Reinforcement Learning** A significant line of research has sought to improve the robustness of policies learned via reinforcement learning. Domain randomization (Tobin et al., 2017), where policies are trained on randomized environments, is a popular approach due to its simplicity and effectiveness. Risk-aware RL methods that try to optimize over tail distributions of uncertainties (Yang et al., 2021; Stachowicz & Levine, 2024) are another approach to improving robustness. Most directly related is RARL (Pinto et al., 2017), which solves a robust RL problem by training an adversarial disturbance policy. This implicitly assumes feasibility under the worst-case disturbance. Our approach does not assume feasibility under the worst-case. Instead, we learn a policy that is robust to the largest possible subset of the parameter space.

**Curriculum Learning and Unsupervised Environment Design** Our method of adjusting the initial state distribution to maximize the safe set is reminiscent of curriculum learning in RL, which also adjusts the initial state distribution, albeit to improve learning efficiency (Narvekar et al., 2020). Methods such as VDS (Zhang et al., 2020b) and PLR (Jiang et al., 2021b) apply different heuristics to prioritize initial states that can provide more informative learning signals. Closely related is unsupervised environment design (UED), which tackles producing a distribution over environments

that "supports the continued learning of a particular policy" (Dennis et al., 2020). Many UED methods choose environments that maximize regret, differing in how regret is estimated and how the selection occurs (Dennis et al., 2020; Jiang et al., 2021a; Lanier et al., 2022; Parker-Holder et al., 2022; Rutherford et al., 2024). While our method also focuses on the initial state distribution, we specifically tackle parameter-robust avoid problems rather than general RL tasks, enabling us to better exploit structure, such as with a feasibility classifier.

**Safe Reinforcement Learning** There is a large body of work on safe RL that adds safety constraints to the original unconstrained MDP objective, whether in the form of constrained MDPs (Altman, 1999) or almost-sure (Sootla et al., 2022) or state-wise (Wachi et al., 2018) safety. This results in a constrained optimization problem where the goal is to optimize the original task objective while satisfying a safety constraint that often conflicts with the task objective. The main challenge in safe RL often lies in how to generalize RL algorithms that perform unconstrained optimization to the constrained setting, and is usually solved with techniques from constrained optimization such as primal-dual optimization with Lagrange multipliers (Ray et al., 2019; Stooke et al., 2020; Li et al., 2021), interior-point methods (Liu et al., 2020), projection-based methods (Yang et al., 2020; Xu et al., 2021), epigraph reformulations (So & Fan, 2023; Zhang et al., 2025), or penalty method (Sootla et al., 2022; Tasse et al., 2023). Our work differs in that we only have a safety objective and no task objective and focus on maximizing the set of parameters that are safe. Consequently, the usual concerns in safe RL of balancing task performance and safety do not arise in our setting.

## 3 PROBLEM FORMULATION

We consider deterministic discrete-time dynamics with parameters $\theta \in \Theta$

$$s_{k+1} = f_\theta(s_k, a), \quad s_0 = s_0(\theta). \tag{1}$$

where $s_k \in \mathcal{S} \subseteq \mathbb{R}^{n_s}$ is the state at time step $k \in \mathbb{N}_{\geq 0}$, $a \in \mathcal{A} \subseteq \mathbb{R}^{n_a}$ is the control, $f_\theta : \mathcal{S} \times \mathcal{A} \to \mathcal{S}$ is the dynamics function under $\theta$, $s_0 : \Theta \to \mathcal{S}$ maps from the parameter space to the initial state, and $\mathcal{S}, \mathcal{A}$ denote state and action spaces respectively.

We first introduce the *parameter-robust* avoid problem, which we illustrate with an example. Consider the case of autonomous driving, where we wish to find a safe policy. Here, safety means avoiding entering unsafe states such as collisions or losing grip when the road is wet or icy. However, instead of only considering a *single* weather condition, we wish for the policy to be safe for *all* weather conditions, including the most adverse ones such as heavy rain or snowfall. Mathematically, we want to find a policy $\pi_\theta \in \mathcal{M} := \{ \pi_\theta \mid \mathcal{S} \times \Theta \to \mathcal{A} \}$ such that for all weather conditions (i.e., parameters) $\theta \in \Theta$, including the most unsafe ones, the system starting from state $s_0(\theta)$ remains outside a set of unsafe (failure) states $\mathcal{F}_\theta \subseteq \mathcal{S}$ for all time [2]. Compared to the standard HJ avoid problem (Mitchell et al., 2005), this allows the dynamics $f_\theta$ and the unsafe set $\mathcal{F}_\theta$ to vary with $\theta$.

Let $h_\theta : \mathcal{S} \to \mathbb{R}$ be a safety function and the unsafe set $\mathcal{F}_\theta := \{ s \mid h_\theta(s) > 0 \}$ be its strict zero-superlevel set, i.e., the set of all unsafe states $s$ under $h_\theta$. Our problem can be transformed into the standard avoid problem using an *augmented* state space $\tilde{\mathcal{S}} := \mathcal{S} \times \Theta$ where $\theta$ of the augmented dynamics is fixed. Then, our problem is equivalent to the following HJ optimal safe control problem:

$$V_{\text{reach}}(s; \theta) := \min_{\pi_\theta \in \mathcal{M}} \max_{k \geq 0} h_\theta(s_k), \quad \text{s.t. } s_{k+1} = f_\theta(s_k, \pi_\theta(s_k)), \tag{2}$$

where the zero-sublevel set of the safety value function [3] $V_{\text{reach}} : \tilde{\mathcal{S}} \to \mathbb{R}$ contains all state-parameter pairs that are feasible, i.e., there exists a policy that renders the system safe for all time (Mitchell et al., 2005). Let $\tilde{\mathcal{S}}_0 := \{ (s_0(\theta), \theta) \mid \theta \in \Theta \}$ denote the set of all initial state-parameter pairs. If all such initial state-parameter pairs are feasible according to the value function $V_{\text{reach}}$, i.e., $\tilde{\mathcal{S}}_0 \subseteq \tilde{\mathcal{S}}_{\text{safe}}$ for the zero-sublevel set $\tilde{\mathcal{S}}_{\text{safe}} := \{ (s_0, \theta) \mid V_{\text{reach}}(s_0; \theta) \leq 0 \}$, then the original *parameter-robust* avoid problem is feasible and the minimizer $\pi_\theta^*$ of (2) also solves the original problem (Section A.1). Note that using state augmentation allows us to treat both initial states and parameters as initial *augmented* states without loss of generality.

---

[2] Equivalently, this is a contextual MDP (Hallak et al., 2015) with deterministic dynamics and an almost-sure safety RL objective (Sootla et al., 2022) where the unsafe states should be avoided with probability 1.

[3] Note that this is not the expected cumulative sum of rewards and hence is **not** the traditional RL value function. However, it does satisfy similar properties such as the *safety Bellman equation* (Fisac et al., 2019).

Although some existing works (Fisac et al., 2019) directly solve (2), we use a different approach that enables the use of existing RL algorithms and avoids the need to come up with new optimizers for the *max-over-time* objective (Veviurko et al., 2024). Let $T_{\theta,\boldsymbol{s}_0} := \min\{\, k \geq 0 \mid h_\theta(\boldsymbol{s}_k) > 0 \,\}$ be the first hitting time of $\mathcal{F}_\theta$, and define the value function $V : \tilde{\mathcal{S}} \to \{\, 0, 1 \,\}$ as

$$V(\boldsymbol{s}_0; \theta) := \max_{\pi_\theta \in \mathcal{M}} J(\pi_\theta, \theta), \ \ J(\pi_\theta, \theta) := \sum_{k=0}^{T_{\theta,\boldsymbol{s}_0}} -\mathbb{1}\{h_\theta(\boldsymbol{s}_k) > 0\}, \ \ \boldsymbol{s}_{k+1} = f_\theta(\boldsymbol{s}_k, \pi_\theta(\boldsymbol{s}_k)). \quad (3)$$

Then, the zero-sublevel sets of $V_{\text{reach}}$ is equivalent to the zero level set of $V$ (see Section A.2), and solving one solves for the other. Now, the *sum-over-time* objective in (3) matches the objective used in standard Markov Decision Processes (MDPs) and can be solved using off-the-shelf methods.

Standard deep RL assumes the initial state is drawn from a distribution $\rho$, but we need to solve $V$ for *all* initial states $(\boldsymbol{s}_0, \theta) \in \tilde{\mathcal{S}}_0$. A naive choice such as a uniform $\rho$ over $S_0$ can be problematic. For example, in autonomous driving, staying safe on icy roads at high speeds is harder than under normal conditions; if weather is sampled uniformly, an RL algorithm will learn more slowly and thus perform worse in snowstorms than on sunny days. We illustrate this on a toy MDP in Section B.

One idea is to instead solve a *robust* problem over the state space $\mathcal{X}$ in a similar spirit to a *static* version of robust reinforcement learning (Morimoto & Doya, 2005; Pinto et al., 2017), i.e.

$$\max_{\pi_{(\cdot)} \in \mathcal{M}} \min_{\theta \in \Theta} \sum_{k=0}^{T_{\theta,\boldsymbol{s}_0}} -\mathbb{1}\{h_\theta(\boldsymbol{s}_k) > 0\}, \quad (4)$$

where an *adversary* learns the worst-case parameter for the current policy during training, and the policy $\pi$ always plays against its most challenging parameter. However, if some $\theta$ makes the objective nonzero for every $\pi_\theta$, then every $\pi_\theta$ attains the same suboptimal value of 1. In a driving task, for example, the adversary could choose an extreme snowstorm and a high initial velocity that guarantees loss of traction. Since no policy can avoid failure in this setting, the optimal response is effectively to do nothing. Yet, training only on such impossible scenarios prevents the policy from learning to handle easier, feasible parameters (e.g., good weather).

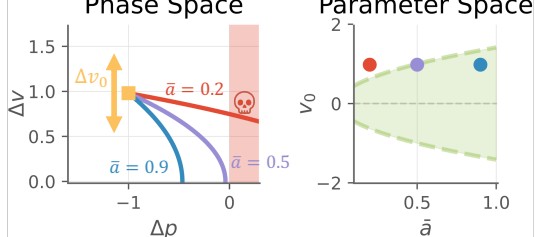

Figure 2: **Adaptive Cruise Control Example.** We illustrate an example using an autonomous driving scenario. The parameters $\theta = [\bar{a}, v_0]$ model varying max acceleration $\bar{a}$ (e.g., from weather) and initial relative velocity $\Delta v_0$. We want to avoid crashing into the car in front ($\Delta p \leq 0$). (**Left**) We plot trajectories for different $\bar{a}$ and (**Right**) visualize the feasible parameter set $\Theta^*$ in green.

This issue can be solved by restricting the maximization to a set of *feasible* parameters $\Theta^* \subseteq \Theta$, but this set is not known a priori. We thus propose the parameter-robust avoid problem with *unknown feasibility* as the following constrained optimization problem

$$\max_{\Theta' \subseteq \Theta} |\Theta'|, \ \text{s.t.} \ \max_{\pi_{(\cdot)} \in \mathcal{M}} \min_{\theta \in \Theta'} \sum_{k=0}^{T_{\theta,\boldsymbol{s}_0}} \mathbb{1}\{h_\theta(\boldsymbol{s}_k) > 0\} = 0, \ \ \boldsymbol{s}_0 = s_0(\theta), \ \ \boldsymbol{s}_{k+1} = f_\theta(\boldsymbol{s}_k, \pi_\theta(\boldsymbol{s}_k)). \quad (5)$$

Unlike the robust setting, we relax the (potentially infeasible) requirement that the *entire* parameter space $\Theta$ is feasible and instead look for the largest feasible subset $\Theta^* \subseteq \Theta$. We illustrate this using an autonomous driving example (Fig. 2), where the parameters $\theta = [\bar{a}, v_0]$ control the max acceleration $\bar{a}$ (e.g., from weather) and initial relative velocity $\Delta v_0$, and safety is defined as not crashing into the car in front ($\Delta p \leq 0$). Only a subset of the full parameter space is feasible, and solving (5) equates to identifying this subset and finding a policy that is safe under such feasible parameters. In the next section, we propose a practical algorithm to solve (5) that simultaneously identifies the feasible set $\Theta^*$ and learns a policy $\pi$ that solves the safe optimal control problem over this set.

## 4 FEASIBILITY-GUIDED EXPLORATION

In this section, we discuss three key challenges in solving the parameter-robust avoid problem with unknown feasibility (5) and how we propose to address them.

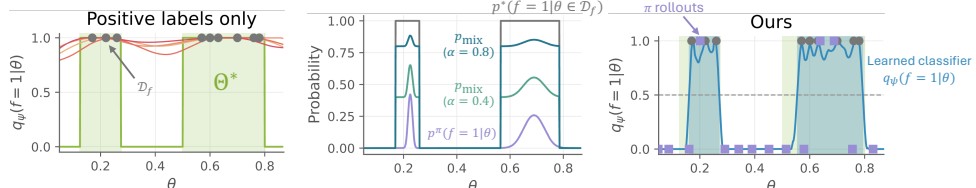

Figure 3: **(Left)** We only have positive labels for feasibility from the set of parameters that were measured to be safe $\mathcal{D}_{\mathfrak{f}}$. Since we don't have negative labels, standard supervised learning cannot be used. **(Center)** We address this by defining a target distribution $p_{\mathrm{mix}}$ via mixing positive labels from $p_{\mathcal{D}_{\mathfrak{f}}}(\theta)$ with noisy labels from $p^\pi$, controlled by $\alpha$. **(Right)** Approximating $p_{\mathrm{mix}}(\mathfrak{f} = 1|\theta)$ with $q_\psi(\mathfrak{f} = 1|\theta)$ using rollout samples and $\mathcal{D}_{\mathfrak{f}}$ samples yields a feasibility classifier with no false positives and a controllable false-negative rate with $\alpha$ and $\rho$ (Theorem 1).

## 4.1 ESTIMATING THE FEASIBLE PARAMETER SET $\Theta^*$

We first tackle the challenge of estimating a feasible set of parameters $\Theta' \subseteq \Theta^*$. This is important: if $\Theta'$ includes infeasible parameters, the robust constraint in (5) becomes unsatisfiable and the policy degenerates. However, feasibility labels are asymmetric: under deterministic dynamics, any parameter observed to be safe is feasible, while an unsafe observation may simply reflect a suboptimal policy. Because only positive labels are reliable, standard supervised learning cannot be applied.

Our goal is to learn a conservative classifier: it should never mark an infeasible parameter as feasible (zero false positives), while keeping false negatives low. To do this, we exploit the label asymmetry by training on two data sources: (i) parameters observed to be safe, which provide trustworthy positive labels, and (ii) on-policy samples, which may include false negatives but reveal the boundary of $\Theta^*$. Mixing these sources yields a classifier that stays conservative despite noisy negative labels. We do this using a data distribution $p_{\mathrm{mix}}$ that upweights reliable positives and downweights potentially noisy negatives. As shown next, this yields a classifier with zero false-positives and a controllable false-negative rate.

Formally, let $\mathfrak{f} = 1$ denote the event that the system remains safe for all time, and let $p^*(\mathfrak{f} = 1|\theta) = \mathbb{1}\{\theta \in \Theta^*\}$ be true probability that $\theta$ is feasible. We approximate $\Theta^*$ by the set of empirically safe parameters $\mathcal{D}_{\mathfrak{f}} := \{\, \theta \mid \exists \pi_\theta, J(\pi_\theta, \theta) = 0 \,\} \subseteq \Theta^*$, and define $p_{\mathcal{D}_{\mathfrak{f}}}(\theta)$ as the uniform distribution over $\mathcal{D}_{\mathfrak{f}}$. Let $p^\pi(\mathfrak{f} = 1 \mid \theta)$ be the safety probability of $\theta$ under policy $\pi$, and let $\rho(\theta)$ be an arbitrary sampling distribution over $\Theta$. For a mixing coefficient $\alpha \in (0, 1)$, we form the mixture (see Fig. 3)

$$p_{\mathrm{mix}}(\mathfrak{f}, \theta) = \alpha\, p^*(\mathfrak{f} \mid \theta)p_{\mathcal{D}_{\mathfrak{f}}}(\theta) + (1 - \alpha)p^\pi(\mathfrak{f} \mid \theta)\rho(\theta). \tag{6}$$

which combines positive samples from $\mathcal{D}_{\mathfrak{f}}$ and samples with noisy labels from $\rho$. We approximate $p_{\mathrm{mix}}$ with $q_\psi : \Theta \to [0, 1]$ by solving the following variational inference problem

$$\min_\psi D_{\mathrm{KL}}\Big(p_{\mathrm{mix}}(\mathfrak{f}, \theta) \,\big\|\, q_\psi(\mathfrak{f}|\theta)p_{\mathrm{mix}}(\theta)\Big). \tag{7}$$

whose optimal solution is the conditional

$$p_{\mathrm{mix}}(\mathfrak{f}|\theta) = \frac{\alpha p^*(\mathfrak{f} \mid \theta)p_{\mathcal{D}_{\mathfrak{f}}}(\theta) + (1 - \alpha)p^\pi(\mathfrak{f} \mid \theta)\rho(\theta)}{\alpha p_{\mathcal{D}_{\mathfrak{f}}}(\theta) + (1 - \alpha)\rho(\theta)}. \tag{8}$$

Thresholding yields the classifier $\phi(\theta) = \mathbb{1}\{q_\psi(\mathfrak{f} = 1 \mid \theta) \geq \beta\}$ for $\beta \in (0, 1)$. If the parameter $\theta \in \Theta^{*\complement}$ is infeasible, then $p_{\mathrm{mix}}(\mathfrak{f} = 1|\theta) = 0$, ensuring perfect classification of infeasible parameters. If the parameter $\theta \in \mathcal{D}_{\mathfrak{f}} \subseteq \Theta^*$ is measured to be feasible, then

$$\theta \in \mathcal{D}_{\mathfrak{f}} \implies p_{\mathrm{mix}}(\mathfrak{f} = 1|\theta) = 1 - \Big(1 - p^\pi(\mathfrak{f} = 1|\theta)\Big)\zeta, \quad \zeta := \frac{1 - \alpha}{\alpha}\frac{\rho(\theta)}{p_{\mathcal{D}_{\mathfrak{f}}}(\theta)} \tag{9}$$

so $\zeta$ controls the attenuation of false negatives (derivation in Section A.6). When $\pi$ is optimal, $p^\pi(\mathfrak{f} = 1 \mid \theta) = 1$ and the error vanishes. Even for suboptimal $\pi$, we can reduce $\zeta$ either by taking $\alpha \to 1$ (making $p_{\mathrm{mix}}$ approach $p^*$) or by choosing $\rho$ to have have low density on $\Theta^*$. Theorem 1 formalizes conditions under which $q_{\psi^*}$ is accurate on $\Theta^*$. Moreover, by choosing $\rho = q_\psi(\theta|\mathfrak{f} = 0)$, we can provide provide guarantees on classifier accuracy that are independent of the policy (Proposition 3).

In practice, we take $\rho$ to be the rollout distribution of $\theta$ under $\pi$, so that $p^\pi(\mathfrak{f} \mid \theta)\rho(\theta)$ corresponds to the joint distribution induced by on-policy rollouts. Since we minimize the forward KL, samples

from $p_{\text{mix}}$ suffice to optimize the objective; these are obtained simply by mixing on-policy samples with positive examples from $\mathcal{D}_{\mathfrak{f}}$.

**Remark 1** (Comparison to Density Modeling Approaches). Prior work models feasibility by estimating a density over $\mathcal{D}_{\mathfrak{f}}$ (Kang et al., 2022; Castaneda et al., 2023), and require choosing a density threshold which is less principled; we compare with density methods in Section 5.2.

## 4.2 FEASIBLE ROBUST OPTIMIZATION WITH SADDLE-POINT FINDING

With an estimate of the feasible parameter set $\Theta^*$ from Section 4.1, we now address the robust optimization problem in (5). While RARL (Pinto et al., 2017) can be used to solve this maximin problem, its Gaussian policy adversary makes it difficult to constraint its output to feasible parameters, and such adversarial dynamics are often unstable in practice (Zhang et al., 2020a).

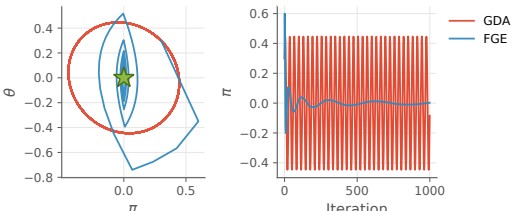

Instead, we adopt saddle-point methods from online learning (Orabona, 2019), which allow the adversary to be explicitly restricted to $\Theta^*$ and provide theoretical convergence guarantees under suitable conditions. We retain the alternating structure between policy updates and selecting worst-case parameters, but compute the adversary's action via *best-response* over the approximated $\Theta^*$. Sublinear regret for the policy is needed for convergence, so we use follow-the-regularized-leader (FTRL) with a quadratic regularizer $\psi$, which keeps successive policies close (Orabona, 2019), stabilizing the policy updates and avoiding oscillations as the worst-case $\theta$ changes. A quick comparison with gradient descent ascent (GDA, Benaïm & Hirsch, 1999), whose iterates RARL approximates with Deep RL, shows the stability from FTRL improves convergence (Fig. 4). The iterates then take the form

Figure 4: **Gradient Descent Ascent (GDA) vs FGE on a Bilinear Game.** On $\max_{\pi \in [-1,1]} \min_{\theta \in [-1,1]} \pi\theta$ for $\pi$, GDA fails to converge in last-iterate, while FGE using (12) converges to the saddle point at the origin. Both converge in average iterate (see Section I). We plot the average $\theta$ for FGE.

$$\theta_{t+1} := \underset{\theta \in \Theta^*}{\arg\min} \ J(\pi_{t+1}, \theta), \quad (10) \qquad \pi_{t+1} := \underset{\pi}{\arg\max} \ -\psi_t(\pi) + \sum_{i=1}^{t} J(\pi, \theta_i). \quad (11)$$

Then, arguments from (Orabona, 2019, Ch. 12) show that (11) has average-iterate convergence to the saddle-point under suitable assumptions (see Section C). We use deep RL to approximate (11) by treating the sum as an expectation and linearizing $J$ around $\pi$ to obtain (see Section A.5 for details):

$$\pi_{t+1} = \pi_t + \eta_t \nabla_\pi \frac{1}{t} \sum_{i=1}^{t} J(\pi, \theta_i) = \pi_t + \eta_t \nabla_\pi \mathbb{E}_{\theta \sim \mathcal{D}_{\theta,t}}[J(\pi, \theta)], \quad \mathcal{D}_{\theta,t} = \text{Uniform}(\{\theta_i\}_{i=1}^{t}). \quad (12)$$

where $\mathcal{D}_{\theta,t}$ is the discrete uniform distribution over the set of maximizers (the *rehearsal buffer*). Using techniques from (Farina et al., 2020), a Monte-Carlo approximation of the expectation in (12) still results in sublinear regret *with high probability* for $\pi$ and hence still converges to the saddle-point *with high probability*. Since (12) performs gradient ascent on $J$, we implement it as a policy update step and apply any on-policy RL algorithm, such as PPO, to the equivalent RL problem.

$$\pi_{t+1} = \underset{\pi}{\arg\max} \ \mathbb{E}_{\theta \sim \mathcal{D}_{\theta,t}}\Big[ \sum_{k=0}^{T_{\theta,s_0}} -\mathbb{1}\{h_\theta(\boldsymbol{s}_k) > 0\}\Big], \ \theta_{t+1} = \underset{\theta^{(i)}}{\arg\min} \ J(\pi_t, \theta^{(i)}), \ \theta^{(i)} \sim p(\cdot | \theta \in \Theta^*). \quad (13)$$

We use the negative indicator as the reward function. Since the exact minimizer is intractable, we sample a batch of $\theta \sim p(\cdot \mid \theta \in \Theta^*)$ use a neural network estimator of $J$ to select an approximate worst-case $\theta_{t+1}$. While the assumptions required for theoretical convergence such as convex-concavity and access to the exact best-response oracles may not be satisfied for the problems we consider, we still observe empirical benefits like improved stability, as shown in our results. These theoretical results provide insight on why our method works well empirically, but theoretically proving convergence is not feasible under current mathematical frameworks and is still an open challenge.

## 4.3 EXPANDING THE FEASIBLE SET ESTIMATE

A remaining challenge is how to expand the measured feasible set $\mathcal{D}_{\mathfrak{f}}$ so that it better approximates the true feasible region $\Theta^*$. We can only control false negatives for parameters within $\mathcal{D}_{\mathfrak{f}}$, and $\mathcal{D}_{\mathfrak{f}}$

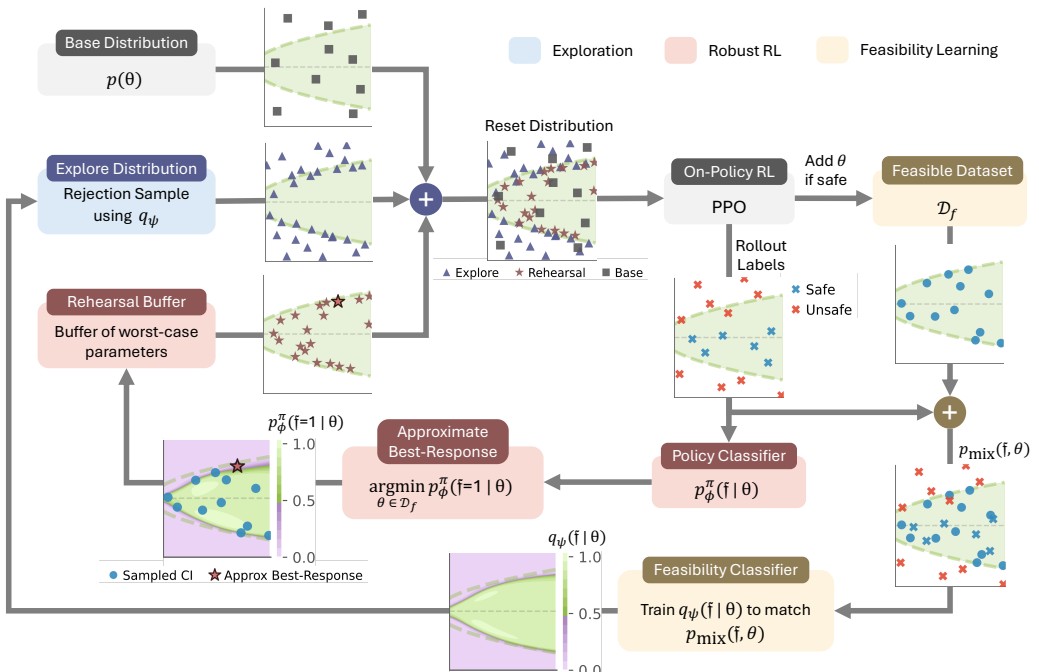

Figure 5: **Feasibility-Guided Exploration (FGE).** Starting from an on-policy RL algorithm, we adapt the initial state distribution as a mixture of the **base** (■), **exploration** (▲) and **rehearsal** (★) distributions. The **exploration** component expands the feasible set and the **rehearsal** buffer targets feasible parameters that the current policy underperforms. After each episode, newly discovered feasible parameters $\theta$ are added to the dataset $\mathcal{D}_{\mathfrak{f}}$. A mixture of samples from $\mathcal{D}_{\mathfrak{f}}$ (●) and the episode (✖, ✖) forms $p_{\mathrm{mix}}(\mathfrak{f}, \theta)$ used to train the feasibility classifier $q_\psi$. The feasibility classifier $q_\psi$ guides rejection sampling for the **explore** distribution. We also train a policy-conditioned classifier $p^\pi$ to predict $\pi$'s performance given $\theta$, enabling approximate best-response selection of worst-case feasible $\theta$ (★) for **rehearsal**. We use the ACC example and plot its parameter space in all plots (see Fig. 2).

only expands when the policy is safe for new parameters during rollouts. This creates a negative feedback loop: a weak policy fails on many parameters, keeping $\mathcal{D}_{\mathfrak{f}}$ small, which in turn restricts the training distribution and prevents improvement on harder parameters.

To break this cycle, we introduce an explicit exploration mechanism that proactively seeks parameters whose feasibility is still uncertain. Using the feasibility classifier $\phi$, we draw candidate parameters that are currently estimated to be infeasible by rejection sampling $\theta \sim p(\cdot \mid \phi(\theta) = 0)$. This encourages the policy to improve on parameters outside the known feasible region, allowing the algorithm to discover new safe parameters and expand $\mathcal{D}_{\mathfrak{f}}$ and thus improve the classifier over time.

### 4.4 FEASIBILITY-GUIDED EXPLORATION

We now present Feasibility-Guided Exploration (FGE; Fig. 5, Algorithm 2). Each iteration, we update the policy using (13), instantiated with PPO, to solve the parameter-robust avoid subproblem under the current feasible set estimate $\tilde{\Theta}^*$. During on-policy rollouts, we augment the rehearsal buffer $\mathcal{D}_{\theta,t}$ by mixing in the exploration distribution (Section 4.3) and the base distribution. The resulting mixture preserves the base distribution's support while concentrating density on difficult yet feasible parameters (from the rehearsal buffer) and on currently infeasible ones (from the explore distribution). Next, we perform the best-response step in (13), where we use a learned policy classifier instead of $J$ (Section D.3.2). Finally, we train the feasible classifier $\phi$ by optimizing $q_\psi$ to match data drawn from both the measured feasible set $\mathcal{D}_{\mathfrak{f}}$ and on-policy rollouts generated by the current policy $\pi$.

### 5 EXPERIMENTS

We compare FGE against baselines from the following categories. **Robust RL:** Domain randomization (DR) (Tobin et al., 2017), the most popular method for robustifying RL policies. We also compare against RARL (Pinto et al., 2017) as a representative work on robust RL. **Curriculum Learning:**

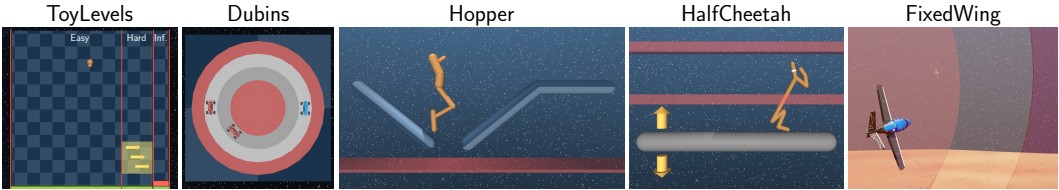

Figure 6: **Benchmark Tasks.** Unsafe regions to be avoided are shown in **red** for each environment.

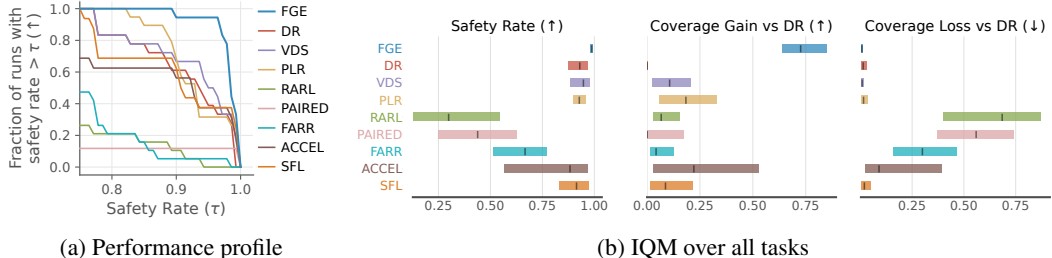

(a) Performance profile             (b) IQM over all tasks

Figure 7: **FGE has the highest safety rates.** (**Left**) Performance profile (Agarwal et al., 2021). At every threshold, FGE has the most runs with higher safety rate than the threshold, dominating all other baseline methods. (**Right**) IQM over all tasks. FGE achieves the highest safety rate by rendering the most hard parameters safe without losing coverage on easy parameters.

VDS (Zhang et al., 2020b) and PLR (Jiang et al., 2021b) are two curriculum learning methods that do not require expert knowledge in designing a curriculum. **UED:** PAIRED (Dennis et al., 2020), FARR (Lanier et al., 2022), ACCEL (Parker-Holder et al., 2022) and SFL (Rutherford et al., 2024) are all recent methods from UED that have been shown to outperform DR. All RL algorithms are instantiated with the PPO algorithm (Schulman et al., 2017) and optimize for (13) but with a different distribution over the parameters $\theta$ that each method defines (see Section G.2 for full details).

**Benchmarks.** We evaluate our method on a suite of tasks consisting of an illustrative example, two MuJoCo environments (Todorov et al., 2012), a driving scenario, and a high-dimensional fixed-wing aircraft simulator (So & Fan, 2023; Heidlauf et al., 2018) (Fig. 6). We also test on a lunar lander from (Matthews et al., 2025) with RGB image-based observations. Details can be found in Section G.3.

**Evaluation Metrics.** As not all parameters $\theta$ are feasible, the highest achievable safety rate for each environment depends on the size of the feasible set for that environment. To enable fair comparison *across* environments, we measure the **safety rate** *conditioned* on the parameter being feasible, where we conservatively estimate that a parameter is feasible if any method can make it safe. Additionally, to understand where the differences arise, we also define **coverage gain** to measure how many "hard" parameters a method can discover as feasible, and **coverage loss** to measure how many "easy" parameters a method cannot discover as feasible, where we use the performance of DR as a proxy for parameter difficulty. Details on these metrics are in Section G.1.

## 5.1 RESULTS

Following evaluation recommendations from Agarwal et al. (2021), we plot the performance profile in Fig. 7a. FGE is the highest curve, indicating that it consistently achieves high safety rate across tasks. Next, to examine differences between methods, we plot the safety rate, coverage gain, and coverage loss (Fig. 7), where FGE achieves the greatest coverage gain relative to DR across all environments. Training curves in Section D.2 show that FGE remains stable during training.

Next, we perform case studies to understand why FGE achieves the highest coverage gains.

**FGE achieves gains in coverage by allocating more samples to rare and difficult parameters.** We first analyze Dubins (Fig. 8) and examine challenging parameters where the ego car must squeeze through a tight gap to remain safe. In particular, we find that parameters where the ego car must *overtake* are extremely rare (bottom left pixel in Fig. 8). Comparing FGE and VDS, we observe that FGE allocates vastly more samples to this condition than VDS. Consequently, FGE remains safe on this rare challenging overtake condition while VDS crashes.

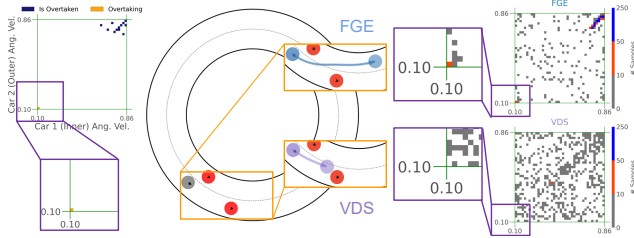

Figure 8: **FGE renders challenging initial conditions safe by allocating more samples to them.**
(**left**) We analyze the Dubins environment and find two types of challenging initial conditions where the ego car must squeeze through a tight gap to remain safe. In particular, the overtake condition is extremely rare. (**right**) FGE allocates vastly more samples to this condition than VDS. (**center**) Consequently, FGE is able to solve this challenging overtake condition while VDS crashes.

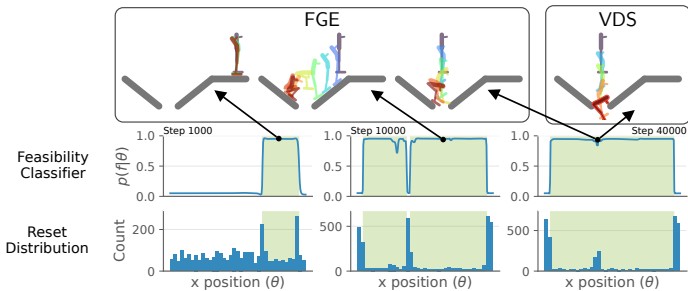

Figure 9: **Rejection Sampling Enables Efficient Exploration in FGE.** The explore distribution in FGE effectively allocates samples to regions that have not been solved yet. By the middle plot, FGE has concentrated a third of its samples on the gap between the two platforms, eventually solving this region by the end of training. In contrast, VDS fails to remain safe in this region.

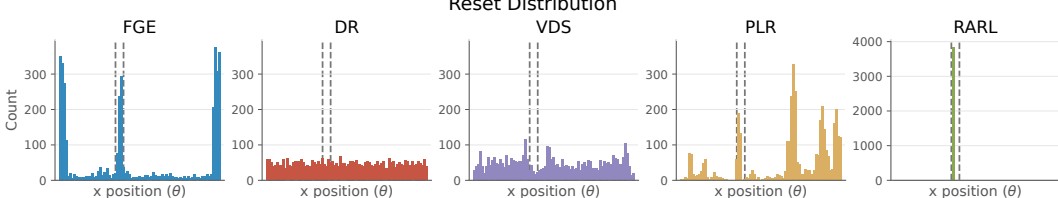

Figure 10: **Heuristics used by existing methods do not concentrate samples as efficiently as FGE.** Only FGE is able to effectively concentrate its samples in the difficult to solve gap region on Hopper.

**The explore distribution in FGE efficiently samples rare and difficult parameters via rejection sampling.** A similar story holds on Hopper. Here, the small gap between platforms requires a recovery maneuver to avoid falling through. We visualize the learned feasibility classifier $\phi$ across training iterations (Fig. 9). Early in training, the robot quickly learns to stay safe on flat ground, and the classifier $\phi$ marks this region as feasible. The explore distribution stops sampling from this solved region and eventually concentrates a third of samples on the gap once all other regions are solved. As in Dubins, this enables FGE to stay safe in the gap while VDS does not.

**Heuristics used by existing methods fail to concentrate samples as efficiently as FGE.** To test our hypothesis that FGE's coverage gains arise from its sampling distribution, we compare reset distributions on Hopper (Fig. 10), marking the gap region with dashed vertical lines. Although methods such as VDS and PLR bias sampling towards certain regions, they do not concentrate samples within the gap as well as FGE. UED methods (PAIRED, FARR, ACCEL, SFL) also fail to produce efficient distributions despite their regret-based motivations, largely due to large regret approximation errors (Rutherford et al., 2024). We compare with UED in more detail in Section E.

**FGE scales to high-dimensional observations and parameters.** We now examine whether FGE scales to high-dimensional observations and parameters by testing on Lander2D and Lander9D, both

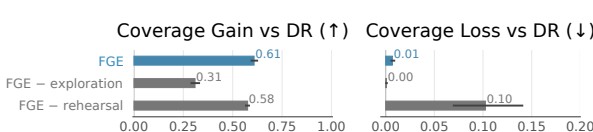

Figure 11: **FGE scales to high-dimensional observations (Lander2D) and high-dimensional parameter spaces (Lander9D).** FGE maintains the highest safety rates even on tasks with RGB image observations (Lander2D, Lander9D) and with 9D parameter spaces (Lander9D).

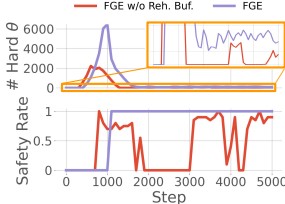

Figure 12: Removing either exploration or rehearsal distributions significantly hurts performance on Dubins.

Figure 13: Removing the rehearsal buffer destabilizes training on ToyLevels.

of which use RGB image observations (Fig. 11). We compare against DR and the two best-performing baseline methods (VDS, PLR). FGE continues to achieving the highest safety rates.

## 5.2 ABLATION STUDIES

Having observed that FGE's gains in coverage come from the effectiveness of the explore distribution, we next answer the following research questions to better understand the components of FGE: **(Q1)** Is the explore distribution responsible for the strong coverage gain of FGE? **(Q2)** How does the rehearsal buffer affect the performance? **(Q3)** How important is the feasibility classifier?

**(Q1) Exploration is essential for high safety rates.** Removing the explore distribution from FGE significantly reduces coverage (Fig. 12). Thus, the explore buffer is essential to FGE's performance.

**(Q2) The rehearsal buffer is critical for stability.** Removing the rehearsal buffer leads to a smaller drop in coverage gain, but higher coverage loss (Fig. 12) as the training becomes unstable (Fig. 13).

**(Q3) The proposed classifier using mixture distributions performs better than density-based methods.** We compare our proposed approach to learning classifier $\phi$ from Section 4.1 to a learned density model using planar flows (PF, Rezende & Mohamed, 2015) and Neural Spline Flows (NSF, Durkan et al., 2019) (Fig. 14). Using a classifier significantly outperforms both PF and NSF, with NSF still achieving higher coverage than the next best-performing baseline (PLR).

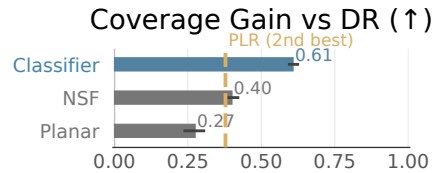

Figure 14: Replacing the feasibility classifier with a density-based classifier hurts performance on Dubins.

## 6 CONCLUSION

We have proposed a novel approach to solving parameter-robust avoid problems with unknown feasibility via simultaneous feasible-set estimation and saddle-point finding and demonstrated the strong performance of our method compared to prior approaches in solving a multitude of challenging benchmark problems, where our method learns policies that achieve significantly larger feasible sets.

**Limitations** Our proposed method approximates the best-response oracles of the saddle-point finding algorithm, thus the theoretical convergence guarantees may not hold. This is common in UED works (e.g., by assuming access to a best-response oracle (Dennis et al., 2020)). These theoretical results provide insight on why our method works well empirically, but theoretically proving convergence is not feasible under current mathematical frameworks and is still an open challenge. Moreover, our framework assumes deterministic dynamics, which enables determining feasibility from a single safe rollout. Handling safety and feasibility for stochastic dynamics via approaches such as chance constraints is more complex for our proposed method compared to techniques such as domain randomization and is a promising future direction.

## 7 REPRODUCIBILITY STATEMENT

All proofs of theoretical results are included in Section A. Implementation details and hyperparameters of all algorithms are included in Section G. We also include the source code of our algorithm in the supplementary materials.

## ACKNOWLEDGEMENT

This material is based upon work supported by the Under Secretary of Defense for Research and Engineering under Air Force Contract No. FA8702-15-D-0001 or FA8702-25-D-B002. Any opinions, findings, conclusions or recommendations expressed in this material are those of the author(s) and do not necessarily reflect the views of the Under Secretary of Defense for Research and Engineering. This work was also supported in part by a grant from Amazon.

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

# A PROOFS

## A.1 RELATIONSHIP BETWEEN THE PARAMETER-ROBUST AVOID PROBLEM AND THE STANDARD REACHABILITY PROBLEM

We define the parameter-robust avoid problem as the following constraint satisfaction problem

$$\min_{\pi_\theta \in \mathcal{M}} \quad 0 \tag{14}$$

$$\text{s.t.} \quad h_\theta(\boldsymbol{s}_k) \leq 0, \quad \forall k \geq 0 \tag{15}$$

$$\boldsymbol{s}_{k+1} = f_\theta(\boldsymbol{s}_k, \pi_\theta(\boldsymbol{s}_k)), \quad \forall k \geq 0 \tag{16}$$

$$\boldsymbol{s}_0 = x_0(\theta), \quad \forall \theta \in \Theta \tag{17}$$

Define the augmented state $\tilde{\boldsymbol{s}} = [\boldsymbol{s}, \theta]$ and the augmented dynamics $\tilde{f}(\tilde{\boldsymbol{s}}, \boldsymbol{a}) = [f(\boldsymbol{s}, \boldsymbol{a}, \theta), \theta]$. Consider the reachability problem (2), i.e.,

$$V_{\text{reach}}(\boldsymbol{s}_0; \theta) = \min_{\pi_\theta \in \mathcal{M}} \max_{k \geq 0} h_\theta(\boldsymbol{s}_k), \quad \text{s.t. } \boldsymbol{s}_{k+1} = f_\theta(\boldsymbol{s}_k, \pi_\theta(\boldsymbol{s}_k)). \tag{2}$$

Let $\mathcal{S}_0 := \{ (x_0(\theta), \theta) \mid \theta \in \Theta \}$ denote the set of all initial state-parameter pairs, and let $\mathcal{S} := \{ (\boldsymbol{s}_0, \theta) \mid V_{\text{reach}}(\boldsymbol{s}_0; \theta) \leq 0 \}$ denote the zero-sublevel set of the value function $V_{\text{reach}}$. Then, the two problems are equivalent in the following sense.

**Proposition 1.** *If $\mathcal{S}_0 \subseteq \mathcal{S}$, then the parameter-robust avoid problem is feasible.*

*Proof.* We prove via contradiction. Suppose that the parameter-robust avoid problem is infeasible. Then, for all policies $\pi_\theta \in \mathcal{M}$, there exists some $k \geq 0$ and some $\theta \in \Theta$ such that $h_\theta(\boldsymbol{s}_k) > 0$. However, since $(x_0(\theta), \theta) \in \mathcal{S}$, (2) implies that there exists some $\pi_\theta^*$ such that $h_\theta(\boldsymbol{s}_k) \leq 0$ for all $k \geq 0$. This is a contradiction. Therefore, the parameter-robust avoid problem is feasible with policy $\pi_\theta^*$. □

## A.2 THE ZERO-SUBLEVEL SETS OF $V_{\text{REACH}}$ IS EQUAL TO THE ZERO LEVEL SET OF $V$

**Lemma 1.** *The zero-sublevel set of the value functions $V_{reach}$ is equal to the zero level set of $V$, i.e.,*

$$\{ (\boldsymbol{s}, \theta) \mid V_{reach}(\boldsymbol{s}; \theta) \leq 0 \} = \{ (\boldsymbol{s}, \theta) \mid V(\boldsymbol{s}; \theta) = 0 \}. \tag{18}$$

*Proof.* We show via double inclusion.

Consider $\boldsymbol{s}_0$ and $\theta$ such that $V_{\text{reach}}(\boldsymbol{s}_0; \theta) \leq 0$. Then, there exists a control policy $\pi_\theta^*$ such that $h_\theta(\boldsymbol{s}_k) \leq 0$ for all $k \geq 0$, and consequently, $\mathbb{1}_{h_\theta(\boldsymbol{s}_k)>0} = 0$ for all $k \geq 0$. Therefore, $V(\boldsymbol{s}_0; \theta) = \sum_{k=0}^{T_\theta} \mathbb{1}_{h_\theta(\boldsymbol{s}_k)>0} = 0$.

Now, consider $\boldsymbol{s}_0$ and $\theta$ such that $V(\boldsymbol{s}_0; \theta) = 0$. Then, $\sum_{k=0}^{T} \mathbb{1}_{h_\theta(\boldsymbol{s}_k)>0} = 0$, implying that $h_\theta(\boldsymbol{s}_k) \leq 0$ for all $k \geq 0$. Therefore, $V_{\text{reach}}(\boldsymbol{s}_0; \theta) = \min_\pi \max_{k \geq 0} h_\theta(\boldsymbol{s}_k) \leq 0$. □

## A.3 UPPER BOUNDING THE FALSE NEGATIVE RATE

We show that the learned classifier $q_{\psi^*}$ for a given threshold $\beta$ can be accurate given conditions on $\alpha$ and the density ratio $\tilde{p}^*/\rho$ over feasible parameters.

**Theorem 1.** *Let $q_{\psi^*}$ denote the solution to (7), and define $\zeta : \Theta \rightarrow [0, \infty)$ as*

$$\zeta(\theta) := \frac{1-\alpha}{\alpha} \frac{\rho(\theta)}{p_{\mathcal{D}_f}(\theta)}. \tag{19}$$

*Then, for a feasible parameter $\theta_f \in \mathcal{D}_f$, we have that $q_{\psi^*}(f = 1 \mid \theta_f) \geq \beta$ for all $\theta_f$ where*

$$p^\pi(f = 1 | \theta_f) \geq \beta - (1-\beta)\zeta(\theta_f)^{-1}. \tag{20}$$

*For infeasible parameter $\theta_n \in \Theta^{*\complement}$, $q_{\psi^*}$ will always correctly classify it with $q_{\psi^*}(f = 1 \mid \theta_n) = 0$.*

*Proof.* For any infeasible parameter $\theta \in \Theta^{*c}$, we have that $p_{\mathcal{D}_{\mathfrak{f}}}(\mathfrak{f} = 1|\theta) = p^{\pi}(\mathfrak{f} = 1|\theta) = 0$. Hence, $p_{\text{mix}}(\mathfrak{f} = 1|\theta) = 0$.

For a feasible parameter $\theta \in D_{\mathfrak{f}}$, we have that $\tilde{p}^*(\mathfrak{f} = 1|\theta) = 1$. Hence,

$$p_{\text{mix}}(\mathfrak{f} = 1|\theta) = \frac{\alpha p_{\mathcal{D}_{\mathfrak{f}}}(\theta) + (1 - \alpha)p^{\pi}(\mathfrak{f} = 1|\theta)\rho(\theta)}{\alpha p_{\mathcal{D}_{\mathfrak{f}}}(\theta) + (1 - \alpha)\rho(\theta)}. \tag{21}$$

For this to be greater than $\beta$, this implies that

$$\alpha p_{\mathcal{D}_{\mathfrak{f}}}(\theta) + (1 - \alpha)p^{\pi}(\mathfrak{f} \geq 1|\theta) \geq \beta\big(\alpha p_{\mathcal{D}_{\mathfrak{f}}}(\theta) + (1 - \alpha)\rho(\theta)\big) \tag{22}$$

$$p^{\pi}(\mathfrak{f} = 1|\theta) \geq \frac{\beta(\alpha p_{\mathcal{D}_{\mathfrak{f}}}(\theta) + (1 - \alpha)\rho(\theta)) - \alpha p_{\mathcal{D}_{\mathfrak{f}}}(\theta)}{(1 - \alpha)\rho(\theta)} \tag{23}$$

$$= \beta - (1 - \beta)\frac{\alpha}{1 - \alpha}\frac{p_{\mathcal{D}_{\mathfrak{f}}}^{\star}(\theta)}{\rho(\theta)} \tag{24}$$

$$= \beta - (1 - \beta)\zeta(\theta)^{-1}. \tag{25}$$

## A.4 BEHAVIOR OF THE CLASSIFIER UNDER REJECTION SAMPLING

We now consider the case where we take $\rho$ as $q_{\psi}(\theta|\mathfrak{f} = 0)$, i.e.,

$$\rho(\theta) = q_{\psi}(\theta|\mathfrak{f} = 0) = \frac{q_{\psi}(\mathfrak{f} = 0|\theta)p(\theta)}{q_{\psi}(\mathfrak{f} = 0)}. \tag{26}$$

This couples the classifier and the parameter sampling distribution. For $\theta \in \mathcal{D}_{\mathfrak{f}} \subseteq \Theta^*$ If we hold the policy $\pi$ fixed and we iteratively update the classifier $q_{\psi}$ and $\rho$, we obtain the following iterates:

$$\rho^{(k)}(\theta) = q^{(k)}(\theta|\mathfrak{f} = 0) = \frac{q^{(k)}(\mathfrak{f} = 0|\theta)p(\theta)}{q^{(k)}(\mathfrak{f} = 0)}, \tag{27}$$

$$q^{(k+1)}(\mathfrak{f} = 0|\theta) = p^{\pi}(\mathfrak{f} = 0|\theta)\frac{(1 - \alpha)\rho^{(k)}(\theta)}{\alpha p_{\mathcal{D}_{\mathfrak{f}}}(\theta) + (1 - \alpha)\rho^{(k)}(\theta)}. \tag{28}$$

We then have the following result about the behavior of these iterates.

**Proposition 2.** *The fixed-point always exists, and is given by*

$$q^{(\infty)}(\mathfrak{f} = 0|\theta) = \max\left(0, p^{\pi}(\mathfrak{f} = 0|\theta) - \alpha\frac{p_{\mathcal{D}_{\mathfrak{f}}}(\theta)p^{\pi}(\mathfrak{f} = 0)}{p(\theta)}\right), \tag{29}$$

*where*

$$p^{\pi}(\mathfrak{f} = 0) = \int p^{\pi}(\mathfrak{f} = 0|\theta)p(\theta)d\theta = \mathbb{E}_{\theta \sim p(\theta)}\left[p^{\pi}(\mathfrak{f} = 0|\theta)\right]. \tag{30}$$

*Proof.* Note that $\frac{(1-\alpha)\rho^{(k)}(\theta)}{\alpha p_{\mathcal{D}_{\mathfrak{f}}}(\theta) + (1-\alpha)\rho^{(k)}(\theta)} \in (0, 1)$, so the error rate of the classifier is always upper-bounded by the policy-induced error rate:

$$q^{(k+1)}(\mathfrak{f} = 0|\theta) \leq p^{\pi}(\mathfrak{f} = 0|\theta). \tag{31}$$

Moreover, note that 0 is absorbing for the iterates: if $q^{(k)}(\mathfrak{f} = 0|\theta) = 0$, then $\rho^{(k)}(\theta) = 0$ and thus $q^{(k+1)}(\mathfrak{f} = 0|\theta) = 0$.

For $\theta$ such that $p^{\pi}(\mathfrak{f} = 0|\theta) > 0$, we can analyze the fixed points of the iterates. The fixed point $q^{(\infty)}$ satisfies

$$q^{(\infty)}(\mathfrak{f} = 0|\theta) = p^{\pi}(\mathfrak{f} = 0|\theta)\frac{(1 - \alpha)\frac{q^{(\infty)}(\mathfrak{f}=0|\theta)p(\theta)}{q^{(\infty)}(\mathfrak{f}=0)}}{\alpha p_{\mathcal{D}_{\mathfrak{f}}}(\theta) + (1 - \alpha)\frac{q^{(\infty)}(\mathfrak{f}=0|\theta)p(\theta)}{q^{(\infty)}(\mathfrak{f}=0)}}. \tag{32}$$

Or,

$$q^{(\infty)}(\mathfrak{f} = 0|\theta)\left(\alpha p_{\mathcal{D}_{\mathfrak{f}}}(\theta) + (1 - \alpha)\frac{q^{(\infty)}(\mathfrak{f} = 0|\theta)p(\theta)}{q^{(\infty)}(\mathfrak{f} = 0)} - (1 - \alpha)p^{\pi}(\mathfrak{f} = 0|\theta)\frac{p(\theta)}{q^{(\infty)}(\mathfrak{f} = 0)}\right) = 0 \tag{33}$$

In other words, at the fixed point, either $q^{(\infty)}(\mathfrak{f} = 0|\theta) = 0$ or

$$\alpha p_{\mathcal{D}_{\mathfrak{f}}}(\theta) + (1 - \alpha)\frac{q^{(\infty)}(\mathfrak{f} = 0|\theta)p(\theta)}{q^{(\infty)}(\mathfrak{f} = 0)} - (1 - \alpha)p^{\pi}(\mathfrak{f} = 0|\theta)\frac{p(\theta)}{q^{(\infty)}(\mathfrak{f} = 0)} = 0. \tag{34}$$

Assume the latter. Then,

$$q^{(\infty)}(\mathfrak{f} = 0|\theta) = p^{\pi}(\mathfrak{f} = 0|\theta)\frac{1}{1 + \frac{\alpha}{1-\alpha}\frac{p_{\mathcal{D}_{\mathfrak{f}}}(\theta)}{\rho^{(\infty)}(\theta)}}, \tag{35}$$

$$= p^{\pi}(\mathfrak{f} = 0|\theta)\frac{1}{1 + \frac{\alpha}{1-\alpha}\frac{p_{\mathcal{D}_{\mathfrak{f}}}(\theta)q^{(\infty)}(\mathfrak{f}=0)}{q^{(\infty)}(\mathfrak{f}=0|\theta)p(\theta)}} \tag{36}$$

Rearranging,

$$q^{(\infty)}(\mathfrak{f} = 0|\theta)\left(1 + \frac{\alpha}{1-\alpha}\frac{p_{\mathcal{D}_{\mathfrak{f}}}(\theta)q^{(\infty)}(\mathfrak{f} = 0)}{q^{(\infty)}(\mathfrak{f} = 0|\theta)p(\theta)}\right) = p^{\pi}(\mathfrak{f} = 0|\theta), \tag{37}$$

$$\iff q^{(\infty)}(\mathfrak{f} = 0|\theta) + \frac{\alpha}{1-\alpha}\frac{p_{\mathcal{D}_{\mathfrak{f}}}(\theta)q^{(\infty)}(\mathfrak{f} = 0)}{p(\theta)} = p^{\pi}(\mathfrak{f} = 0|\theta), \tag{38}$$

$$\iff q^{(\infty)}(\mathfrak{f} = 0|\theta) = p^{\pi}(\mathfrak{f} = 0|\theta) - \frac{\alpha}{1-\alpha}\frac{p_{\mathcal{D}_{\mathfrak{f}}}(\theta)q^{(\infty)}(\mathfrak{f} = 0)}{p(\theta)}. \tag{39}$$

Substituting $q^{(\infty)}(\mathfrak{f} = 0|\theta)$ into $q^{(\infty)}(\mathfrak{f} = 0)$, we get

$$q^{(\infty)}(\mathfrak{f} = 0) = \int q^{(\infty)}(\mathfrak{f} = 0|\theta)p(\theta)d\theta, \tag{40}$$

$$= \int \left(p^{\pi}(\mathfrak{f} = 0|\theta) - \frac{\alpha}{1-\alpha}\frac{p_{\mathcal{D}_{\mathfrak{f}}}(\theta)q^{(\infty)}(\mathfrak{f} = 0)}{p(\theta)}\right)p(\theta)d\theta, \tag{41}$$

$$= \int p^{\pi}(\mathfrak{f} = 0|\theta)p(\theta)d\theta - \frac{\alpha}{1-\alpha}q^{(\infty)}(\mathfrak{f} = 0)\int p_{\mathcal{D}_{\mathfrak{f}}}(\theta)d\theta, \tag{42}$$

$$= p^{\pi}(\mathfrak{f} = 0) - \frac{\alpha}{1-\alpha}q^{(\infty)}(\mathfrak{f} = 0). \tag{43}$$

Hence,

$$q^{(\infty)}(\mathfrak{f} = 0) = p^{\pi}(\mathfrak{f} = 0) - \frac{\alpha}{1-\alpha}q^{(\infty)}(\mathfrak{f} = 0), \tag{44}$$

$$\implies q^{(\infty)}(\mathfrak{f} = 0) = (1 - \alpha)p^{\pi}(\mathfrak{f} = 0). \tag{45}$$

Substituting back, we get the fixed point

$$q^{(\infty)}(\mathfrak{f} = 0|\theta) = p^{\pi}(\mathfrak{f} = 0|\theta) - \alpha\frac{p_{\mathcal{D}_{\mathfrak{f}}}(\theta)p^{\pi}(\mathfrak{f} = 0)}{p(\theta)}. \tag{46}$$

Since $q^{(\infty)}(\mathfrak{f} = 0|\theta) \geq 0$, we have that

$$q^{(\infty)}(\mathfrak{f} = 0|\theta) = \max\left(0, p^{\pi}(\mathfrak{f} = 0|\theta) - \alpha\frac{p_{\mathcal{D}_{\mathfrak{f}}}(\theta)p^{\pi}(\mathfrak{f} = 0)}{p(\theta)}\right). \tag{47}$$

$\square$

Consequently, we can use this to bound classifier accuracy even under the worst-case policy $\pi$ where $p^{\pi}(\mathfrak{f} = 0|\theta) = 1$:

**Proposition 3.** *Under the worst-case policy where $p^{\pi}(\mathfrak{f} = 0|\theta) = 1$, for desired error $\epsilon \in (0, 1)$,*

$$\alpha \geq (1 - \epsilon)\frac{p(\theta)}{p_{\mathcal{D}_{\mathfrak{f}}}(\theta)} \implies q^{(\infty)}(\mathfrak{f} = 0|\theta) \leq \epsilon. \tag{48}$$

*Proof.* If $q^{(\infty)}(\mathfrak{f} = 0|\theta)$ is zero, then it is trivially satisfied. Suppose it is not. Then,

$$q^{(\infty)}(\mathfrak{f} = 0|\theta) = 1 - \alpha \frac{p_{\mathcal{D}_\mathfrak{f}}(\theta)}{p(\theta)} \le \epsilon \tag{49}$$

$$\iff \alpha \ge (1 - \epsilon) \frac{p(\theta)}{p_{\mathcal{D}_\mathfrak{f}}(\theta)}. \tag{50}$$

$\square$

## A.5 DERIVATION OF (12)

Using a quadratic proximal regularizer $\psi_t(\pi) = \frac{t}{2\eta} \|\pi - \pi_t\|^2$ and linearized losses $J(\pi) \approx J(\pi_t) + \nabla J(\pi_t)^\top (\pi - \pi_t)$, then taking the gradient and setting it to zero, we obtain

$$-\frac{t}{\eta}(\pi - \pi_t) + \sum_{i=1}^{t} \nabla J(\pi_t) = 0 \tag{51}$$

$$\implies \pi = \pi_t + \eta \frac{1}{t} \sum_{i=1}^{t} \nabla J(\pi_t) \tag{52}$$

## A.6 DERIVATION OF (9)

$$\theta \in \mathcal{D}_\mathfrak{f} \implies p_{\mathrm{mix}}(\mathfrak{f} = 1|\theta) = 1 - \Big(1 - p^\pi(\mathfrak{f} = 1|\theta)\Big) \underbrace{\Big(1 + \frac{\alpha}{1-\alpha} \frac{p_{\mathcal{D}_\mathfrak{f}}(\theta)}{\rho(\theta)}\Big)^{-1}}_{:=\zeta}, \tag{53}$$

For $\theta \in \mathcal{D}_\mathfrak{f}$, starting from the definition of $p_{\mathrm{mix}}(\mathfrak{f} = 1|\theta)$ in (8),

$$p_{\mathrm{mix}}(\mathfrak{f} = 1|\theta) = \frac{\alpha p^*(\mathfrak{f} = 1 \mid \theta) p_{\mathcal{D}_\mathfrak{f}}(\theta) + (1-\alpha) p^\pi(\mathfrak{f} = 1 \mid \theta)\rho(\theta)}{\alpha p_{\mathcal{D}_\mathfrak{f}}(\theta) + (1-\alpha)\rho(\theta)}, \tag{54}$$

$$= \frac{\alpha p_{\mathcal{D}_\mathfrak{f}}(\theta) + (1-\alpha) p^\pi(\mathfrak{f} = 1 \mid \theta)\rho(\theta)}{\alpha p_{\mathcal{D}_\mathfrak{f}}(\theta) + (1-\alpha)\rho(\theta)}, \tag{55}$$

$$\tag{56}$$

$\square$

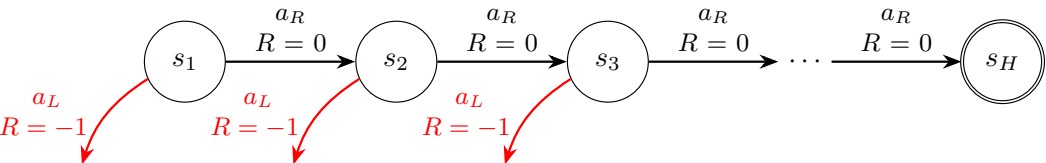

Figure 15: Chain MDP

# B  FURTHER ANALYSIS OF CHAIN MDP

## B.1  (SINGLE) CHAIN MDP ANALYSIS

We first define the single-chain MDP of length $H$, which consisting of $H$ states $s_1, \ldots, s_H$, two actions $a_L$ and $a_R$, and initial state $s_1$. For any state $i$, taking the action $a_L$ terminates the episode and returns a reward of $-1$, while taking the action $a_R$ moves to the next state $s_{i+1}$. The final state $s_H$ is a terminal state, which returns a reward of $0$ and terminates the episode. We visualize the single-chain MDP in Fig. 15.

We first analyze the single-chain MDP. For simplicity, we parameterize the policy by a scalar $\pi \in \mathbb{R}$, where we take action $a_R$ with probability $\pi$ and action $a_L$ with probability $1 - \pi$.

The single-trajectory REINFORCE update law is

$$\nabla_\pi V^\pi(s_1) \approx \sum_t \nabla_\pi \log \pi(a_t|s_t) R_{\geq t}. \tag{57}$$

where $a_t, s_t, R_{\geq t}$ are random variables. Let $T$ denote the (random) time step where the episode terminates by taking $a_L$ if $T < H$, and let $T = H$ denote the episode terminating by reaching the terminal state $s_H$.

We also have $\nabla \log_\pi \pi(a = a_R) = \frac{1}{\phi}$ and $\nabla \log_\pi \pi(a = a_L) = -\frac{1}{1-\phi}$.

Also,

$$R_{\geq t} = \begin{cases} -\gamma^{T-t}, & t < T, T \neq H, \\ -1, & t = T, T \neq H, \\ 0, & T = H. \end{cases} \tag{58}$$

Note that the entire trajectory is captured by the random-variable $T$. Hence, it is sufficient to compute the gradient conditioned on each value of $T$, then take the expectation over $T$. Let $g(T)$ denote the gradient conditioned on $T$.

For $T < H$, we have

$$g(T) = \sum_{t=1}^{T-1} \nabla_\pi \log \pi(a = a_R) R_{\geq t} + \nabla_\pi \log \pi(a = a_L) R_{\geq T} \tag{59}$$

$$= \sum_{t=1}^{T-1} \frac{1}{\pi}(-\gamma^{T-t}) + (-\frac{1}{1-\pi})(-1) \tag{60}$$

$$= \frac{-1}{\pi} \frac{\gamma - \gamma^T}{1-\gamma} + \frac{1}{1-\pi}. \tag{61}$$

If $T = H$, then $g(H) = 0$ since the return is $0$.

Finally, looking at the distribution of $T$, we have

$$P(T = t) = \pi^{t-1}(1-\pi), \quad 1 \leq t < H, \tag{62}$$

$$P(T = H) = \pi^{H-1}. \tag{63}$$

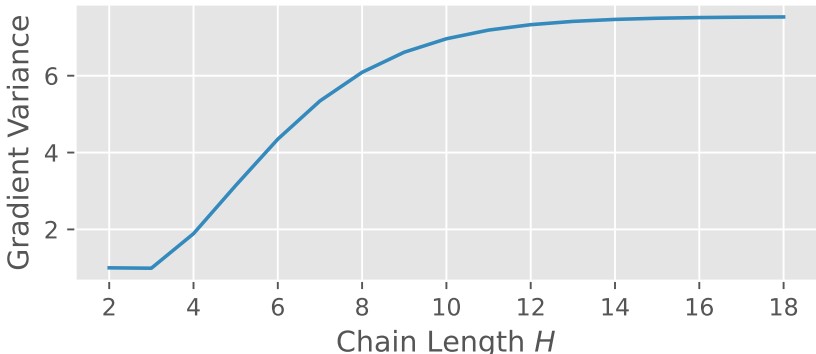

Figure 16: Variance of the single-sample gradient estimator for the chain MDP.

Hence,

$$\mathbb{E}[g(T)] = \sum_{t=1}^{H-1} P(T = t)g(t), \operatorname{Var}[g(T)] = \sum_{t=1}^{H-1} P(T = t)(g(t) - \mathbb{E}[g(T)])^2. \qquad (64)$$

Plotting the variance numerically in Fig. 16 for $\pi = 0.5$ and $\gamma = 0.99$, we see that the variance increases with $H$.

## B.2 MULTI-CHAIN MDP ANALYSIS

We now consider the multi-chain MDP, which consists of $P$ independent chains of length $H_i$ for $i = 1, \ldots, P$. The parameter $\theta \in \{1, \ldots, P\}$ controls which chain's initial state to start from. See Fig. 17 for a visualization.

Here, we assume the policy is parametrized by $\pi^i$ for $i = 1, \ldots, P$, i.e., each chain has a separate policy. Suppose we sample $\theta$ from the uniform distribution over $\{1, \ldots, P\}$. Let $V(H)$ denote the variance of a single-chain MDP with horizon $H$. Then, the variance of the resulting multi-sample monte-carlo REINFORCE gradient estimator will be scaled down equally across all $P$ chains, i.e., if each of the $P$ chains gets a budget of $N$ samples, we obtain a variance of $V(H_i)/N$ for the $i$th chain. Given that larger variances of the gradient estimator theoretically slow down the convergence rate of SGD by reducing the maximum learning rate that can be used Bottou et al. (2018), this means that the policy for the chain with the longest horizon $H_i$ converges the slowest as it has the largest variance.

Instead, if we sample $\theta$ from a distribution proportional to the gradient variance, i.e.,

$$p(\theta_i) \propto V(H_i), \qquad (65)$$

prioritizing chains with larger gradient variances, then the variance will be equal for all chains, with the worst chain converging faster than in the previous case.

Figure 17: Multi Chain MDP

## C CONVERGENCE GUARANTEES TO SADDLE POINTS

We present a proof based on Orabona (2019) that a combination of best-response for the $\theta$ player and follow-the-regularized-leader for the $\mu$ player results in convergence to the saddle-point under a suitable set of assumptions. For analysis, we flip the sign on $J$ to better match the convention from the online learning literature (Orabona, 2019) and consider the minimization objective.

Consider the following algorithm:

---
**Algorithm 1** Saddle-point finding with best-response and follow-the-regularized-leader
---
**Require:** $\mu_1 \in \mathcal{M}$
1: **for** $t = 1, \ldots, T$ **do**
2:     Set $\theta_t \in \arg\max_{\theta \in \Theta} J(\mu_t, \theta)$
3:     Set $\mu_{t+1} = \psi_t(\mu) + \arg\min_{\mu \in \mathcal{M}} \sum_{s=1}^{t} J(\mu, \theta_s)$
4: **end for return** $\theta$
---

where we define the average iterates $\bar{\theta}_T$ and $\bar{\mu}_T$ as

$$\bar{\mu}_T := \frac{1}{T} \sum_{t=1}^{T} \mu_t, \tag{66}$$

$$\bar{\theta}_T := \frac{1}{T} \sum_{t=1}^{T} \theta_t. \tag{67}$$

We now make the following assumptions:

**Assumption 1.** $J : \mathcal{M} \times \Theta \to \mathbb{R}$ *is convex in the first argument and concave in the second argument. Moreover, for any $\theta$, $J(\cdot, \theta)$ is L-Lipschitz, i.e.,*

$$|J(\mu_1, \theta) - J(\mu_2, \theta)| \leq L\|\mu_1 - \mu_2\|. \tag{68}$$

We then have the following result.

**Lemma 2** (Adapted from Orabona (2019)). *Let $J : \mathcal{M} \times \Theta \to \mathbb{R}$ be convex in the first argument and concave in the second argument. Then, we have*

$$0 \leq \sup_{\theta \in \Theta} J(\bar{\mu}_T, \theta) - \min_{\mu \in \mathcal{M}} J(\mu, \bar{\theta}_T) \leq \frac{1}{T} \text{Regret}_T^{\mathcal{M}}(\mu_T'), \tag{69}$$

*for any $\mu'_T \in \arg\min_{\mu \in \mathcal{M}} J(\mu, \bar{\theta})$, where*

$$\text{Regret}_T^{\mathcal{M}}(\mu) := \sum_{t=1}^{T} J(\mu_t, \theta_t) - \sum_{t=1}^{T} J(\mu, \theta_t). \tag{70}$$

*Proof.* By definition of $\text{Regret}_T^{\mathcal{M}}$, for any $\mu \in \mathcal{M}$,

$$\frac{1}{T} \sum_{t=1}^{T} J(\mu_t, \theta_t) - \frac{1}{T} \sum_{t=1}^{T} J(\mu, \theta_t) = \frac{1}{T} \text{Regret}_T^{\mathcal{M}}(\mu) \tag{71}$$

Moreover, since $\theta_t$ is the best-response to $\mu_t$, for any $\theta \in \Theta$, we have that $J(\mu_t, \theta) \leq J(\mu_t, \theta_t)$ for any $t$. Hence,

$$\frac{1}{T} \sum_{t=1}^{T} J(\mu_t, \theta) - \frac{1}{T} \sum_{t=1}^{T} J(\mu_t, \theta_t) \leq 0. \tag{72}$$

Summing (71) and (72) then yields

$$\frac{1}{T} \sum_{t=1}^{T} J(\mu_t, \theta) - \frac{1}{T} \sum_{t=1}^{T} J(\mu, \theta_t) = \frac{1}{T} \text{Regret}_T^{\mathcal{M}}(\mu). \tag{73}$$

Now, Jensen's inequality gives us that, for any $\theta \in \Theta$ and $\mu \in \mathcal{M}$,

$$J(\bar{\mu}_T, \theta) - J(\mu, \bar{\theta}_T) \leq \frac{1}{T} \sum_{t=1}^{T} J(\mu_t, \theta) - \frac{1}{T} \sum_{t=1}^{T} J(\mu, \theta_t). \tag{74}$$

Combining this with (73) then gives

$$J(\bar{\mu}_T, \theta) - J(\mu, \bar{\theta}_T) \leq \frac{1}{T} \text{Regret}_T^{\mathcal{M}}(\mu). \tag{75}$$

Choosing $\mu = \mu'_T \in \arg\min_{\mu \in \mathcal{M}} J(\mu, \bar{\theta}_T)$ and taking suprema over $\theta \in \Theta$ then gives (69), where the non-negativity of the duality gap follows from $\sup_{\theta \in \Theta} J(\mu', \theta) \geq J(\mu', \theta') \geq \inf_{\mu \in \mathcal{M}} J(\mu, \theta')$ for all $\mu'$ and $\theta'$. $\square$

Consequently, if we can show that the regret of the $\mu$ player is sublinear, Lemma 2 implies that the average iterates $(\bar{\mu}_T, \bar{\theta}_T)$ converge to zero duality gap, i.e., a saddle point of the game. We show that this holds next in our case where $\mu$ uses a follow-the-regularized-leader strategy.

**Lemma 3.** *Let $\psi : \mathcal{M} \to \mathbb{R}$ be a closed, $m$-strongly convex function, and define the sequence of regularizers $\psi_t$ as*

$$\psi_t(\mu) = \frac{1}{\eta_{t-1}} \left( \psi(\mu) - \min_z \psi(z) \right), \tag{76}$$

*with the non-increasing sequence $\eta_t$ defined as*

$$\eta_{t-1} = \frac{\alpha\sqrt{m}}{L\sqrt{t}}, \quad t = 1, \ldots, T. \tag{77}$$

*Then, the regret for the $\mu$ player is sublinear, i.e.,*

$$\text{Regret}_T^{\mathcal{M}}(\mu'_T) = O(\sqrt{T}). \tag{78}$$

*Proof.* Under Assumption 1, the conditions for (Orabona, 2019, Corollary 7.7) are satisfied with $\ell_t(\mu) = J(\mu, \theta_t)$ which yields the result. $\square$

Combining these two then gives us the desired convergence result.

**Corollary 1.** *The average iterates $(\bar{\mu}_T, \bar{\theta}_T)$ converge to a saddle point of the game.*

*Proof.* By Lemma 3, the regret for $\mu$ is sublinear. Invoking Lemma 2 then gives us the convergence of the average iterates $(\bar{\mu}_T, \bar{\theta}_T)$ to a saddle point of the game. ☐

## D ADDITIONAL RESULTS AND ABLATIONS

In this section, we include additional results and ablation studies that are not in the main paper due to space constraints.

### D.1 PER-ENVIRONMENT COVERAGE PLOTS

While Fig. 7 provides a summary of the performance of each method across all environments, we now present a per-environment breakdown of the performance of each method in Fig. 18. The conclusion remains the same, with FGE yielding significant increases in coverage gain compared to the next best baseline method. Interestingly, the next best method varies a lot across the different environments: RARL achieves the second-best coverage gain in three out of the five environments, but consistently achieves the worst coverage loss against DR.

### D.2 TRAINING CURVES

We plot the training curves for all environments in Fig. 19.

### D.3 ADDITIONAL ABLATION STUDIES

We next perform additional ablation studies to understand the impact of the design choices of FGE that were not covered in the main paper.

#### D.3.1 DOES A MORE TRADITIONAL EXPLORATION TECHNIQUE SUCH AS RND SOLVE OUR PROBLEM?

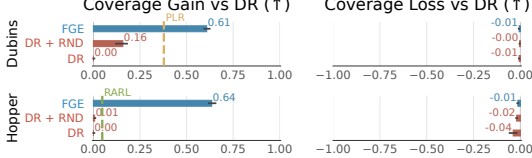

Figure 20: **RND ablation.**

**No, RND is beneficial but does not replace FGE's gains.** In Fig. 20, we add RND Burda et al. (2018) intrinsic reward to DR. We observe slight improvements on coverage gain for DR relative to its non-RND counterpart. However, adding RND to DR neither outperforms FGE nor the next best-performing baseline. Thus, we conclude that exploration techniques such as RND cannot replace FGE in solving the parameter-robust avoid problem with unknown feasibility.

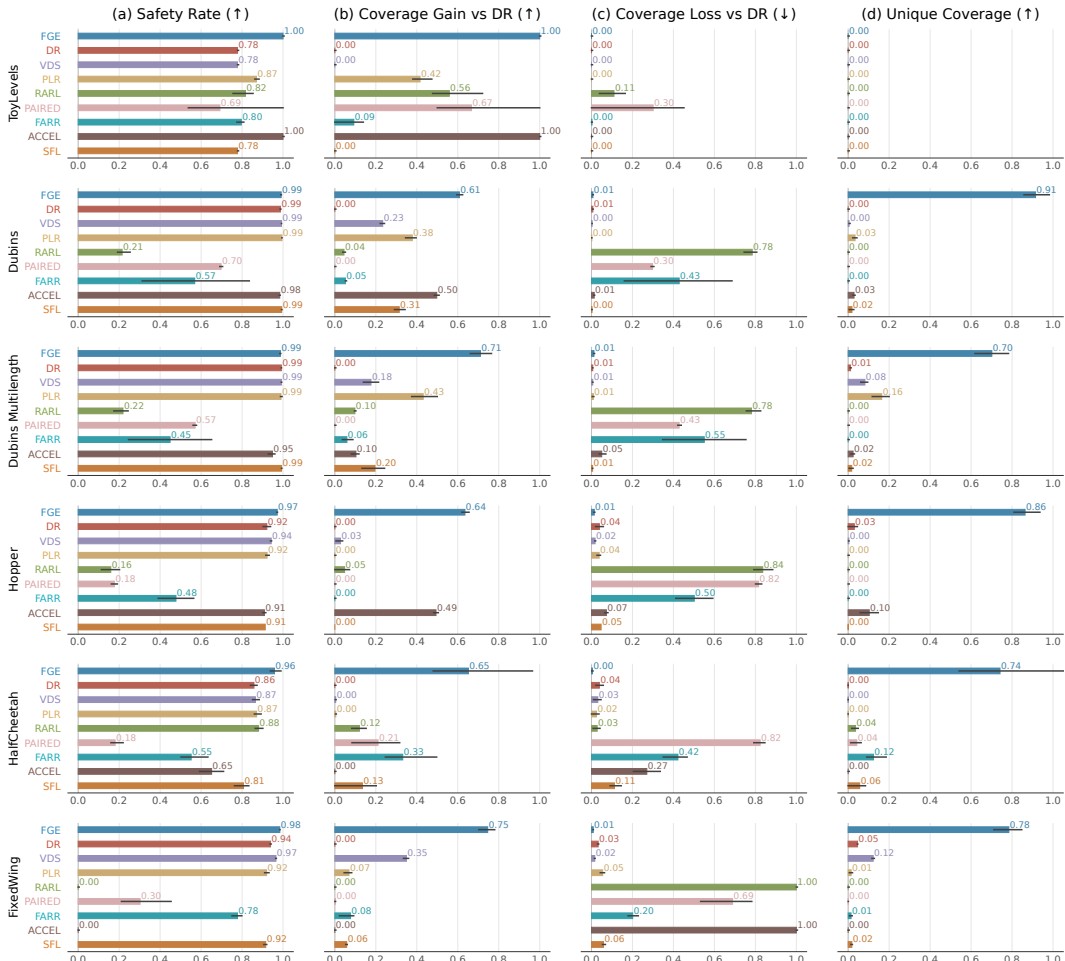

Figure 18: **Coverage Comparison. (a)** FGE achieves near 100% coverage across all environments, and consistently achieves the highest coverage in the harder environments. **(b)** FGE achieves the largest coverage increases relative to DR across all environments, substantially outperforming all other baselines. Other methods can expand on the feasible set of DR, but underperform FGE by 40% to 90% on the harder environments. **(c)** Most baselines do not lose coverage when compared to DR with the exception of RARL due to its instability during training. ACCEL also performs poorly due to the need for each rollout to last for the maximum episode steps, reducing the number of total updates steps taken. **(d)** With the exception of ToyLevels, almost all the unique safe parameters occur due to FGE, suggesting that FGE consistently renders parameters feasible that no other method can.

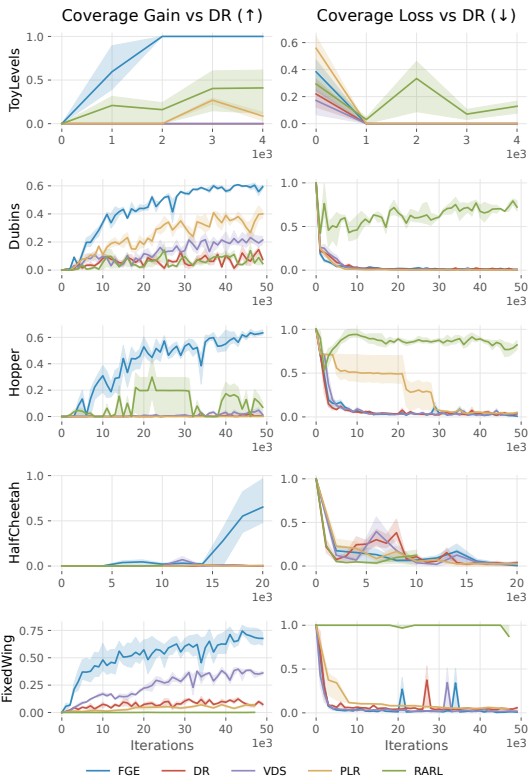

Figure 19: **Training Curves.**

### D.3.2 DOES USING A VALUE FUNCTION OVER A CLASSIFIER MAKE A DIFFERENCE?

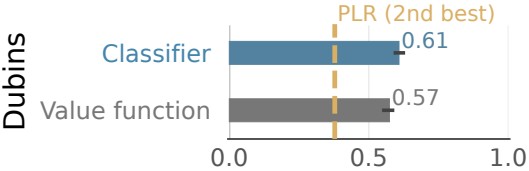

Figure 21: **Value vs classifier for feasibility on Dubins.**

**Both classifier and value function performed similarly, but it should still matter in practice.**
We compare using our policy classifier with an RL value function where if $s \in A_\theta$, $V(s) > 0$, else
$V(s) = 0$. We empirically find on Dubins that using either method does not matter much. However,
we hypothesize that using the RL value function as a policy classifier should incur a few issues in
practice. For instance, if an unsafe state has a long termination sequence, the cumulative discounted
value of that state can become arbitrarily small. This may lead to numerical imprecision that is
expected with any learning-based approximation of the value function, potentially leading to incorrect
classifications.

### D.3.3 HOW SENSITIVE IS OUR CLASSIFIER TO THE MIX FRACTION $\alpha$?

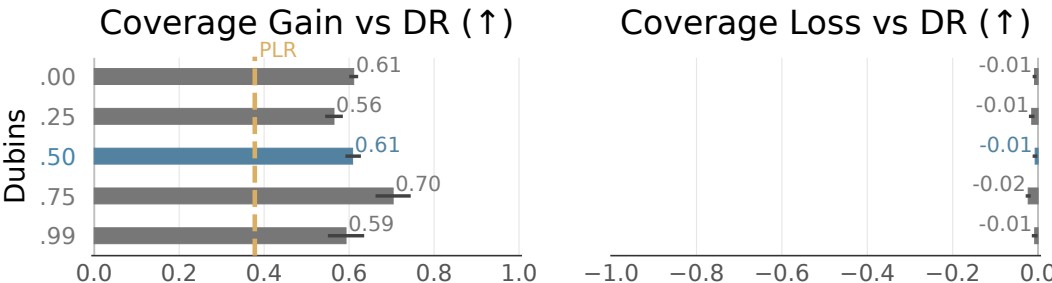

Figure 22: $\alpha$ **sensitivity ablation.**

**It's fairly robust.** Seeing how sensitive NSF's performance is to our choice of threshold from Section 5.2, we conduct a sensitivity test on the mix fraction $\alpha$ for our classifier. From Fig. 22, we observe that $\alpha = 0.75$ performed the best on Dubins. However, from our experiments we did notice that the performance with respect to $\alpha$ differed by task. In addition, FGE still outperforms the next best-performing baseline, PLR, for all values of $\alpha$, indicating that our proposed method's gains are robust to these hyperparameters. For consistency, we default to $\alpha = 0.5$ across all experiments.

## D.4 SAFE REINFORCEMENT LEARNING COMPARISONS

FGE is orthogonal to safe RL methods since it only defines the initial state distribution. Therefore, FGE can be easily paired with any on-policy method, including those in safe RL, to improve safety. We illustrate this by defining a safe RL task on ToyLevels (Section G.3) where the policy must minimize cumulative discounted sum of control input magnitudes subject to safety constraints, i.e.,

$$\min_{\pi} \mathbb{E} \left[ \sum_t ||\pi(\cdot)||_2^2 \right] \tag{79}$$

$$\text{s.t.} \quad \max_{\theta} J(\pi, \theta) = 0 \tag{80}$$

We pair two safe RL algorithms, PPO-lagrangian (PPO-L) and SauteRL(SauteRL, Sootla et al. (2022)), with FGE and the domain randomization baseline (DR). Since both algorithms have a hyperparameter (lagrange multiplier for PPO-L, unsafe penalty for SauteRL), we run a hyperparameter scan for each algorithm and plot the reward vs coverage of each run in Fig. 23.

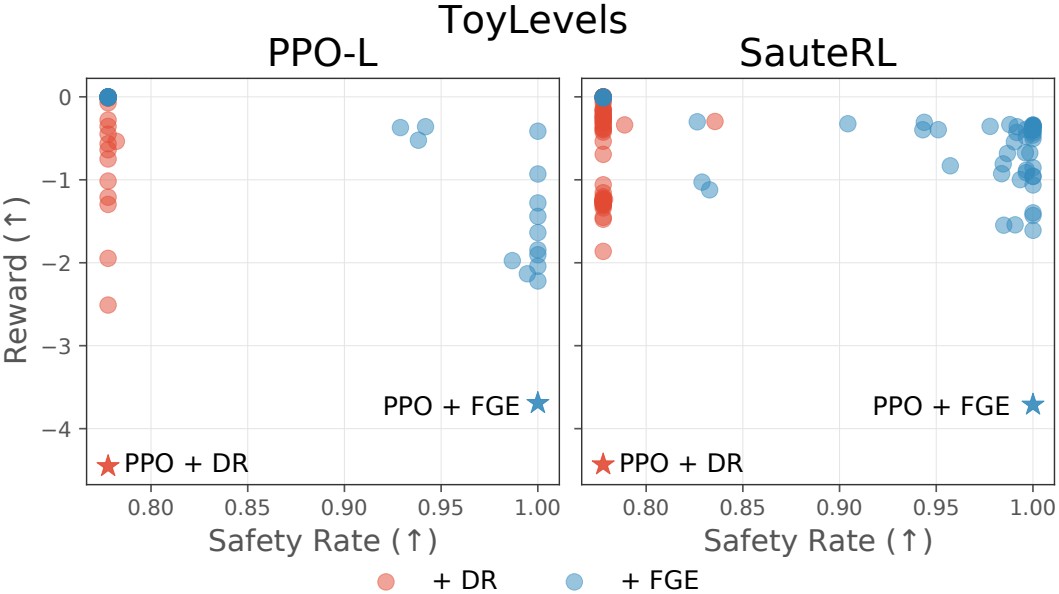

Figure 23: **Safe RL Pareto Frontier.**

We observe that FGE greatly expands the pareto frontier compared to DR in both PPO-L and SauteRL. In addition, all safe RL variants (circles) achieve higher reward than the non-safe RL baselines (stars) as expected, since the non-safe RL methods do not consider the reward. This shows that FGE can be effectively used to complement existing safe RL methods to improve safety over a wider set of parameters.

# E  RELATIONSHIP WITH UNSUPERVISED ENVIRONMENT DESIGN (UED)

Many methods from UED aim to choose parameters that have high regret (Dennis et al., 2020; Lanier et al., 2022; Parker-Holder et al., 2022; Rutherford et al., 2024), with regret for a parameter $\theta$ defined as

$$\text{Regret}(\pi_\theta, \theta) := J(\pi_\theta, \theta) - \max_{\pi_\theta^*} J(\pi_\theta^*, \theta). \tag{81}$$

The challenge with (81) is that the second term involves an optimization problem and is thus intractable. Consequently, UED methods propose different methods of approximating this term, which we briefly discuss below.

**PAIRED (Dennis et al., 2020):**  In PAIRED, regret is approximated by training an *antagonist* policy that aims to maximize the regret. Specifically, a separate PPO policy is used, with per-timestep rewards replaced by per-episode rewards computed as the difference between the return of $\pi_\theta$ and the antagonist policy. However, one issue is that the antagonist policy will also initially struggle for parameters $\theta$ that are hard but feasible, resulting in both policies being unsafe and resulting in a regret of $0$. This does not yield any learning signal to the antagonist policy and results in a bad approximation of the regret.

**FARR (Lanier et al., 2022):**  FARR approximates the regret directly by using PPO to solve for the optimal policy $\pi_\theta^*$ for *each* parameter $\theta$ for which the regret is evaluated on. This is computationally expensive for tasks where converging requires a large number of environment interactions. Moreover, in cases where the parameter $\theta$ is very challenging, it is possible that PPO returns a suboptimal solution, especially if learning to be safe for $\theta$ normally requires some type of curriculum learning or generalization of knowledge from easier parameters. Hence, in most realistic scenarios where there are constraints on computational resources, the quality of the regret approximation directly trades off with the number of parameters $\theta$ that can be evaluated.

**ACCEL (Parker-Holder et al., 2022):**  ACCEL approximates the regret using the positive value loss (Rutherford et al., 2024). However, as noted by Rutherford et al. (2024), this is unable to reliably

identify the frontier of learning and often prioritizes parameters that are already solved. The cause is hypothesized by Rutherford et al. (2024) to stem from inaccurate value estimation. Moreover, in contrast to typical PPO implementations that only have a short rollout length before updating the policy and value function, every rollout of ACCEL has the maximum episode length [4]. This requirement drastically reduces the number of policy updates that can happen for a fixed budget of environment interactions.

**SFL (Rutherford et al., 2024):** Instead of using the regret, SFL computes a *learnability* metric, defined as $p(1-p)$ for the success rate $p$. However, this means that the learnability is set to 0 for parameters that are currently not successful, which is problematic for difficult parameters that are still feasible.

Interestingly, FGE can be interpreted from the lens of UED's regret analysis. Since we tackle reachability problems with deterministic dynamics, we know the value of the problematic second term in (81) for any parameters $\theta$ that were observed to be safe at *any* point during training. Consequently, the feasibility classifier we learn in Section 4.1 can be viewed as a privileged approximation of the second term in (81). Specifically, we take it to be 0 for all parameters predicted to be feasible by the classifier $\phi$ and 1 otherwise. Hence, FGE's best-response procedure in Section 4.2 can be seen as a form of sample-based regret maximization over $\theta$.

## F  FEASIBILITY GUIDED EXPLORATION PSEUDOCODE

We provide the pseudocode for FGE in Algorithm 2.

---

**Algorithm 2** Feasibility-Guided Exploration (FGE)

---

1: Initialize the rehearsal buffer $\mathcal{D}_{\theta,0} = \{\}$
2: Initialize the feasible buffer $\mathcal{D}_{\mathfrak{f}} = \{\}$
3: **for** $t = 0, 1, 2, \ldots$ **do**
4:     Collect a set of trajectories using the current policy but with a mixture of reset distributions (Algorithm 3)
5:     Update the policy and value function with any on-policy RL algorithm (e.g., PPO)
6:     **for** each completed trajectory **do**
7:         **if** Trajectory is feasible **then**
8:             Add the parameter $\theta$ to $\mathcal{D}_{\mathfrak{f}}$ if it is not already in $\mathcal{D}_{\mathfrak{f}}$
9:         **end if**
10:     **end for**
11:     Train the feasibility classifier $q_\psi(\mathfrak{f}|\theta)$ on a mix of $\mathcal{D}_{\mathfrak{f}}$ and the most recent rollouts
12:     Use the most recent rollouts to update the policy classifier $p_\phi^\pi(\mathfrak{f}|\theta)$
13:     Compute the best-response $\theta^* = \arg\min_{\theta \in \mathcal{D}_{\mathfrak{f}}} p_\phi^\pi(\mathfrak{f}|\theta)$
14:     Update the rehearsal buffer with $\mathcal{D}_{\theta,t+1} \leftarrow \mathcal{D}_{\theta,t} \cup \{\theta^*\}$
15: **end for**

---

---

[4] https://github.com/DramaCow/jaxued/blob/0f8f1284677375b889e4f13a32c9617cd009f8c4/examples/maze_plr.py#L102

---

**Algorithm 3** Reset Distribution for FGE

---

**Require:** $p_{\text{base}}$ the probability of sampling $\theta$ from the base distribution $p(\theta)$
**Require:** $p_{\text{explore}}$, the probability of sampling $\theta$ from the explore distribution
**Require:** $p_{\text{rehearse}}$, the probability of sampling $\theta$ uniformly from the rehearsal buffer $\mathcal{D}_{\theta,t}$
**Require:** $p_{\text{rehearse}} + p_{\text{base}} + p_{\text{explore}} = 1$
 1: Let $u \sim \mathcal{U}(0,1)$
 2: **if** $0 \leq u < p_{\text{base}}$ **then**
 3:     **return** $\theta \sim p(\theta)$
 4: **else if** $0 \leq u - p_{\text{base}} < p_{\text{explore}}$ **then**
 5:     **return** $\theta \sim p(\theta | q_\psi(\mathfrak{f} = 1|\theta) < \beta)$
 6: **else**
 7:     **return** $\theta \sim \mathcal{D}_{\theta,t}$ uniformly
 8: **end if**

---

# G EXPERIMENT DETAILS

## G.1 DEFINITION OF METRICS

In this subsection, we describe how the coverage metrics we plot are computed.

Let $\mathfrak{s}_m$ be a binary random variable that is 1 if method $m$ is safe for the (random) parameter $\theta$ and 0 otherwise. When computing the metrics, we assume $\theta \sim p(\theta)$ is sampled from the base distribution. For example, given a random $\theta$, $\mathfrak{s}_{\text{DR}} = 1$ if DR is safe for $\theta$ and 0 otherwise.

Let $\mathfrak{s}_{\text{DR}}^{\cup}$ denote the binary random variable that is 1 if DR is safe for $\theta$ for any seed.

Also, let $\mathfrak{s}_{\text{any}}$ denote the binary random variable that is 1 if, given a random $\theta$, *any* method (including all ablations), over any seed, can keep $\theta$ safe.

We compute the **Safety Rate**, **Coverage Gain vs DR**, **Coverage Loss vs DR** and **Unique Coverage** as conditional probabilities, which we describe in detail next.

**Safety Rate** This is the normal safety rate, conditioned on if a parameter is estimated to be feasible with $\mathfrak{s}_a = 1$.

$$\text{SafetyRate}_m = p(\mathfrak{s}_m = 1 | \mathfrak{s}_{\text{any}} = 1). \tag{82}$$

In other words, if there exists at least one method where parameter $\theta$ is safe, then $\theta$ is for sure feasible and we include it in the feasible set.

**Coverage Gain vs DR** This coverage metric tracks the increase in coverage over $\theta$ for some method $m$ over DR. Let $\text{CovGain}_m$ denote the coverage gain vs DR for method $m$. Then,

$$\text{CovGain}_m = p(\mathfrak{s}_m = 1 | \mathfrak{s}_{\text{any}} = 1, \mathfrak{s}_{\text{DR}}^{\cup} = 0). \tag{83}$$

In other words, we consider the parameters $\theta$ that are safe by method $m$ and are safe by at least one method, but DR is unable to make safe for any seed. Higher coverage gains are better. A value of 1 means that $\mathfrak{s}_m = \mathfrak{s}_{\text{any}}$, i.e., any $\theta$ that is safe by at least one method is also safe by $m$. A value of 0 means that $m$ does not render any additional $\theta$ safe compared to DR, union over all seeds.

**Coverage Loss vs DR** This coverage metric tracks the loss in coverage over $\theta$ for some method $m$ compared to DR. Let $\text{CovLoss}_m$ denote the coverage loss vs DR for method $m$. Then,

$$\text{CovLoss}_m = p(\mathfrak{s}_m = 0 | \mathfrak{s}_{\text{DR}}^{\cup} = 1). \tag{84}$$

In other words, we consider the parameters $\theta$ that are safe by DR for at least one seed, but is unsafe for method $m$. A value of 1 means that $\mathfrak{s}_{\text{DR}}^{\cup} = 1 \implies \mathfrak{s}_m = 0$, i.e., all $\theta$ that are safe by at least one seed of DR are unsafe under method $m$. Lower coverage losses are better. A value of 0 means that any $\theta$ that is safe for DR for at least one seed, method $m$ is also safe.

**Unique Coverage**  This coverage metric tracks how well method $m$ can render parameters $\theta$ safe that no other method can render safe. For a given plot, let $\mathcal{M}$ denote the set of methods we are plotting. Let $\text{UniqueCov}_m$ denote the unique coverage for method $m$. Then,

$$\overline{\text{UniqueCov}}_m = p\left(\mathfrak{s}_m = 1, \max_{m' \in \mathcal{M}, m' \neq m} \mathfrak{s}_{m'} = 0\right), \tag{85}$$

$$\text{UniqueCov}_m = \frac{\overline{\text{UniqueCov}}_m}{\sum_{m'} \overline{\text{UniqueCov}}_{m'}} \tag{86}$$

A value of 1 means that the set of safe parameters for method $m$ is a superset of the safe set for all other methods. A value of 0 means that it is a subset of the union of the safe sets for all other methods.

For all plots, we plot the error bars with the $50\%$ percentile error, i.e., the black bar represents the .25 and .75 quantiles over 3 seeds.

## G.2  DETAILS ON BASELINE METHODS

In this subsection, we provide more details on how we implement each of the baseline methods used in the experiment section.

**All** methods solve (13) and use the same indicator function reward function defined in (13), i.e.,

$$r_\theta(\boldsymbol{s}) = -\mathbb{1}\{h_\theta(\boldsymbol{s}) > 0\}, \tag{87}$$

and terminate when unsafe (i.e., $h_\theta(\boldsymbol{s}) > 0$), differing only in the reset distribution over the parameters. Specifically, let $\rho(\theta)$ denote the reset distribution such that the (13) is written as

$$\max_\pi \; \mathbb{E}_{\theta \sim \rho}\Big[ \sum_{k=0}^{T_{\theta,s_0}} -\mathbb{1}\{h_\theta(\boldsymbol{s}_k) > 0\}\Big]. \tag{88}$$

Then, each baseline method only changes the choice of $\rho$, which can potentially change over training.

**DR (Tobin et al., 2017)**  We take $\rho$ to be the uniform distribution over $\Theta$.

**VDS (Zhang et al., 2020b)**  We train an ensemble of RL value functions, and take $\rho$ to be the probability distribution proportional to the standard deviation, or "disagreement", among the value function predictions.

**PLR (Jiang et al., 2021b)**  We take the $\theta$ to be the "level" in (Jiang et al., 2021b) and use their scheme for choosing the level as $\rho$.

**RARL (Pinto et al., 2017)**  While the adversary in RARL outputs an action each timestep, we construct a variant where the adversary solves a one-step RL problem and only outputs the parameter $\theta$ at the initial timestep.

**UED baselines (PAIRED (Dennis et al., 2020), FARR (Lanier et al., 2022), ACCEL (Parker-Holder et al., 2022), SFL (Rutherford et al., 2024))**  We follow their setup exactly as our problem maps exactly to the PAIRED problem, with our parameters $\theta$ matching up with the UED "environment" or "level" $\theta$.

## G.3  ENVIRONMENT DETAILS

We provide more details for each of the environments visualized in Fig. 6.

**ToyLevels.**

In the ToyLevels environment (Fig. 24), the agent spawns at the top of the map and falls down at a constant rate. The state space $\mathcal{S} \subseteq \mathbb{R}^2$ is composed of the coordinate of the agent. The control space $\mathcal{A} := \{0, 1, 2\}$ corresponds to a set of three controls: shift left by a constant $c$, shift right by

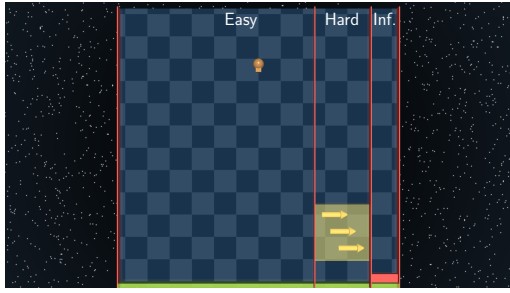

Figure 24: **ToyLevels.**

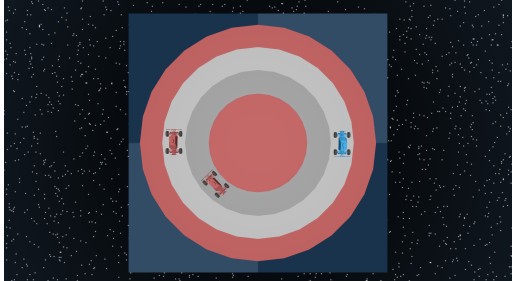

Figure 25: **Dubins.**

$c$, or do not shift. The goal of the agent is to reach the green region at the bottom of the map while avoiding the red regions. We split the spawn states at the top of the map into three levels: Easy, Hard, and Infeasible. The easy region is spacious, and the agent is likely to reach the bottom even with suboptimal controls. The infeasible region is impossible to reach the bottom because there is an obstacle in the way. The hard region is narrow and contains a disturbance region that pushes the agent toward the right at a constant rate. We consider this hard because the agent must learn to move to the left of the disturbance, then output a constant sequence of "shift left" actions in order to reach the bottom safely.

We define $\theta \in \mathbb{R}$ as the x-position of the agent's initial position, and set the base parameter distribution $p(\theta)$ such that the agent resets uniformly in the easy region $90\%$, hard region $0.01\%$, and infeasible region $9.99\%$ of the time. Intuitively, the hard region represents a set of initial conditions that are extremely rare and requires more samples to solve, and the learned policy is slightly different from that requires for other initial states.

**Dubins.** The Dubins environment (Fig. 25) requires an ego car to drive around a two-lane circular track. There are two other cars driving at constant angular velocities, and the ego car must avoid colliding into either of them. The static parameter space $\theta \in \mathbb{R}^2$ is defined by the other cars' angular velocities. We set the base parameter distribution $p(\theta)$ to be uniform in $\left[\frac{\pi}{32}, \frac{11\pi}{40}\right]^2$.

Given the environment circular track's inner radius $r_{\text{inner}}$ and lane width $w_{\text{lane}}$, the outer radius is computed

$$r_{\text{outer}} = r_{\text{inner}} + 2w_{\text{lane}} + 2\epsilon$$

where $\epsilon$ is a small offset applied between the vehicle and each lane boundary. Both ego and other cars are modeled as discs with radius $v_{\text{car}}$. The ego car is initialized on the inner lane on the right side of the track, and other cars are initialized at staggered angular positions across different lanes starting from the left side of the track.

The state space $\mathcal{S} \subseteq \mathbb{R}^4$ is a vector representing the x-coordinate, y-coordinate, heading, and linear velocity. The control space $\left[a_{\text{steer}}, a_{\text{acc}}\right] \in \mathcal{A} \subseteq \mathbb{R}^2$ is the steering and acceleration. Angular velocity $\omega$ of each agent is computed

$$\omega = a_{\text{steer}} \cdot \frac{2v}{(r_{\text{inner}} + r_{\text{outer}})/2}$$

and the ego velocity is updated as

$$v \leftarrow \text{clip}(v + a_{\text{acc}} \cdot dt, v_{\text{min}}, v_{\text{max}})$$

**Hopper.** The Hopper uses the MuJoCo mjx simulator Todorov et al. (2012) and uses the Hopper model from the Deepmind control suite Tassa et al. (2018) (Fig. 26). For this environment, the state space $\mathcal{S} \subseteq \mathbb{R}^{14}$ consists of the joint positions and velocities, while the control space $\mathcal{A} \subseteq \mathbb{R}^4$ controls the torques for each of the four joints. We increase the gear ratios from $[30, 40, 30, 10]$ to $[30, 50, 50, 100]$ to enable the Hopper robot to perform more acrobatic maneuvers such as backflips.

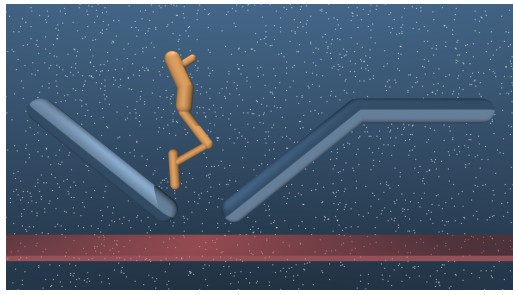

Figure 26: **Hopper.**

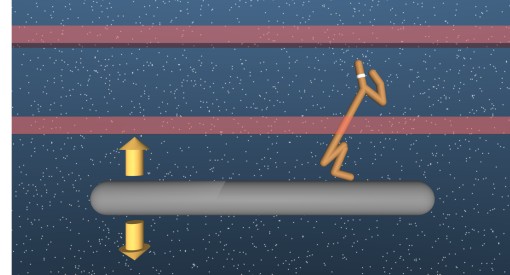

Figure 27: **Cheetah.**

Note that this is *different* from the hopper in OpenAI gym Brockman et al. (2016) as it has four controllable joints instead of three and has a nose.

We additionally add two floating platforms (in blue in Fig. 26). Safety is defined as the $z$-position of the hopper staying above a height of $0.9$, as well as all parts of the hopper except the foot (e.g., thigh, pelvis, torso, nose) not colliding with the two floating platforms.

The parameter $\theta \in \mathbb{R}$ controls the initial x-position of the hopper.

**Cheetah.** In Cheetah also uses the MuJoCo mjx simulator Todorov et al. (2012) and uses the Cheetah model from the Deepmind control suite Tassa et al. (2018). In this environment, the cheetah must keep its head (visualized with a white circle in its head) between the red boundaries (Fig. 27). It sits on a floating platform (grey) that moves up and down at regular intervals. If the platform moves down far enough, the cheetah must stand on its hind legs to remain safe, as shown in the Fig. 27.

For this environment, the state space $\mathcal{S} \subseteq \mathbb{R}^{18}$ consists of the joint positions and velocities, while the control space $\mathcal{A} \subseteq \mathbb{R}^{6}$ controls the torques for each of the six joints.

The parameter $\theta \in \mathbb{R}$ controls the force applied to the moving platform.

**FixedWing.** In FixedWing, the goal is to perform the extended-trail exercise used to practice formation flying for fixed-wing airplanes (Team RV (2002)) (Fig. 28). Specifically, there are two aircraft named *lead* and *chase*. We let the lead aircraft fly in a circular trajectory. The goal of the chase aircraft, which is to be controlled, is to stay within the *extended-trail cone*, which is defined with the following specifications:

- The distance between the airplanes should be within $500\,\mathrm{ft}$ and $1000\,\mathrm{ft}$.

- The chase airplane should be between $30°$ to $45°$ degrees of the lead aircraft, measured in the lead aircraft's frame

These two specifications result in the extended-trail cone taking the shape of the outer shell of a frustrum.

We use the jax model of the aircraft from So & Fan (2023); Heidlauf et al. (2018) which is based off of Morelli (1995); Stevens & Lewis (2004). The state space $\mathcal{S} \subseteq \mathbb{R}^{36}$ composing of the state-space of the two aircraft each with dimension $\mathbb{R}^{16}$ and the guidance controller of the lead aircraft with dimension $\mathbb{R}^{4}$. The control space $\mathcal{A} \subseteq \mathbb{R}^{3}$ consists of the stick commands for the chase aircraft. Importantly, the throttle for the chase aircraft is set to a higher value ($0.4$) than the lead aircraft ($0.28$), which will result in the chase aircraft overtaking (and exiting the extended-trail cone) if the additional energy is not handled properly.

**Lander2D.** In Lander2D, the goal is for the lander in green to avoid colliding with the red circular obstacles, leaving the boundaries of the screen or running out of fuel, and is implemented using Kinetix (Matthews et al., 2025). Consequently, the only way for the lander to survive is to land on the blue platform and stay there without falling off. The parameter $\theta = [x_0, y_0] \in \mathbb{R}^{2}$ describes the initial position of the lander.

We use RGB images for the observation space with a dimension of $64 \times 64 \times 3$ and apply a frame stacking of 3. Moreover, we include a single scalar observation of the remaining fuel.

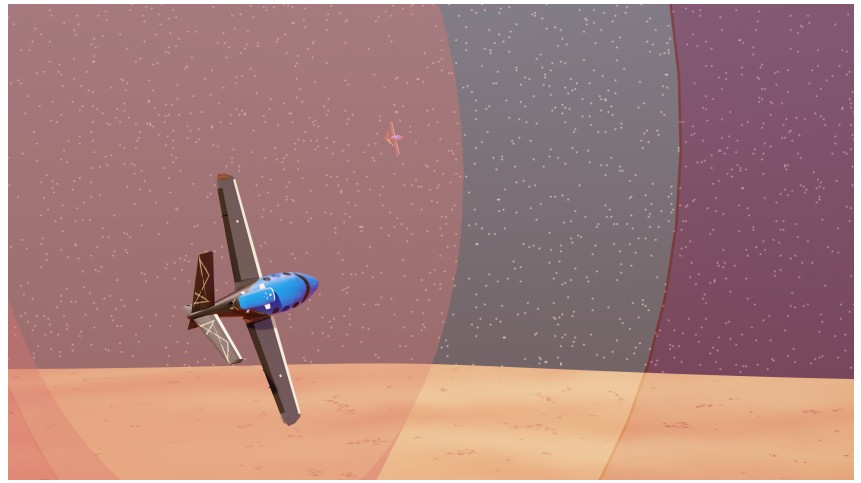

Figure 28: **FixedWing.**

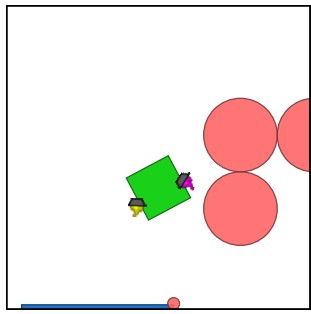

Figure 29: **Lander2D.**

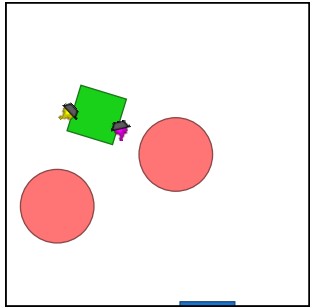

Figure 30: **Lander9D.**

Table 1: Common hyperparameters across all tasks.

| Method | Hyperparameter | Value |
|---|---|---|
| **All** | Neural Network Size | [256, 256] |
| | Activation Function | $\tanh$ |
| | NN Optimizer | Adam |
| | # envs (per collect) | 1024 |
| | # rollout steps | 30 |
| FGE | $p_{\text{base}}$ | 0.02 |
| | $p_{\text{explore}}$ | 0.88 |
| | $p_{\text{rehearse}}$ | 0.1 |
| | Classifier mix fraction $\alpha$ | 0.5 |
| | Classifier threshold $\beta$ | 0.5 |
| PLR | Prioritization | rank |
| | Temperature $\beta$ | 0.1 |
| | Staleness Coefficient $\rho$ | 0.1 |
| VDS | # ensemble | 3 |
| RARL | # policy iterations per round | 50 |
| | # adversary iterations per round | 50 |
| PAIRED | # adversary parameters sampled per iteration | 32 |
| FARR | # PPO steps for best response computation | 50,000 |
| | # Parameters added per iteration | 3 |
| ACCEL | Prioritization | rank |
| | Temperature $\beta$ | 0.1 |
| | Staleness Coefficient $\rho$ | 0.1 |
| | Top K | 4 |
| | Mutate Scale | 0.04 |
| SFL | Update Period $T$ | 50 |
| | Buffer Size $K$ | 100 |
| | Sample Ratio $\rho$ | 0.9 |

**Lander9D.** Lander9D is a harder version of Lander2D, where the parameter $\theta = [x_0, y_0, \theta_0, \text{platform position}, \text{platform width}, \text{obs0}_x, \text{obs0}_y, \text{obs1}_x, \text{obs1}_y] \in \mathbb{R}^9$ describe the positions of the lander, the blue platform and the red obstacles. Everything else remains the same as in Lander2D.

## G.4 HYPERPARAMETERS

Task-independent hyperparameters can be found in Table 1 and task-dependent hyperparameters can be found in Table 2.

## G.5 COMPUTE RESOURCES

All training was performed on a desktop machine with a Intel i7-13700KF CPU and a NVIDIA GeForce RTX 4090 graphics card.

Table 2: Task-specific hyperparameters for each method.

| Method | Hyperparameter | ToyLevels | Dubins | Hopper | Cheetah | FixedWing |
|--------|----------------|-----------|--------|--------|---------|-----------|
| **All** | Policy learning rate | 4e-4 | 5e-5 | 4e-4 | 4e-4 | 4e-4 |
| | Value learning rate | 1e-3 | 8e-4 | 8e-4 | 8e-4 | 8e-4 |
| | # PPO epochs | 5000 | 50000 | 50000 | 50000 | 50000 |
| | Truncate timesteps | 199 | 1000 | 1000 | 1000 | 1000 |
| | Batch size | 30 | 30 | 15 | 30 | 30 |
| | Entropy coefficient | 1e-3 | 5e-3 | 2e-3 | 1e-4 | 1e-4 |

## H  SENSITIVITY OF SADDLE-POINT FINDING TO VIOLATED ASSUMPTIONS

The assumptions required for the convergence-guarantees in Section 4.2 to hold, namely a convex-concave objective function and exact best-response oracles, do not hold in our empirical experiments in Section 5. Nevertheless, we still see empirical benefits from using such a scheme.

A rigorous theoretical treatment of saddle-point finding in the nonconvex-nonconcave setting is out of scope for this work. However, as a preliminary investigation into this phenomenon, we investigate empirically how sensitive the convergence is when we break the necessary assumptions.

To do so, we consider the following minimax optimization problem

$$\min_{\pi \in [-1,1]} \max_{\theta \in [-1,1]} J(\pi, \theta), \tag{89}$$

where the function $J : \mathbb{R} \to \mathbb{R} \to \mathbb{R}$ is defined by

$$J(\pi, \theta) = \mathbb{1}(h(\pi, \theta) > 0), \tag{90}$$

$$h(\pi, \theta) = -1296\pi^4 + 36\pi^2 - 216\pi^2\theta + 2592\pi^3\theta - 4 \tag{91}$$

In this example, the solution to (89) is given by values of $(\pi, \theta)$ inside the red region in Fig. 31a. For any value of $\pi$ within this region, all values of $\theta$ give $J(\pi, \theta) = 0$. Since $J(\pi, \theta) \in \{0, 1\}$, this means that this is the optimal value, i.e., no other choice of $\pi$ results in a lower value of $\max_{\theta \in [-1,1]} J(\pi, \theta)$..

Note that $J$ is not convex-concave, and thus violates Assumption 1. Furthermore, we violate the assumption of best-response oracles by approximating the best-response with random samples (as done in FGE for the $\theta$ maximization), and control the approximation error by controlling the number of random samples.

Using these approximations, we run the iterates defined by (11) and plot the results in Fig. 31 but without using the regularizer $\psi$. First, despite the function not being convex-concave, the saddle-point finding algorithm still manages to converge to the saddle-point and never diverges no matter how coarse the best-response oracle approximation is. Second, as expected, the number of iterations required to converge decreases as the best-response oracle becomes more accurate.

We leave a proper theoretical treatment of whether there do exist relaxed assumptions (e.g., nonconvex-nonconcave and inexact best-reponse oracle) as future work.

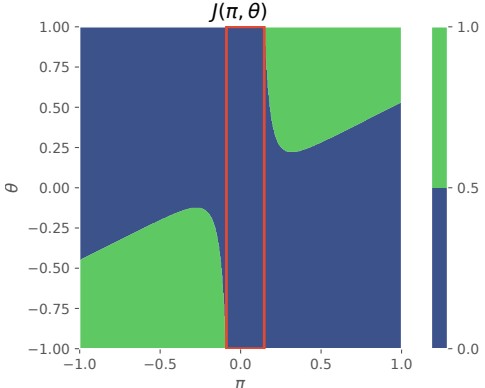
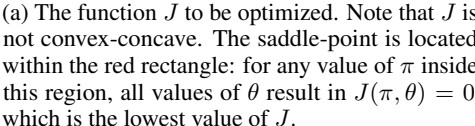
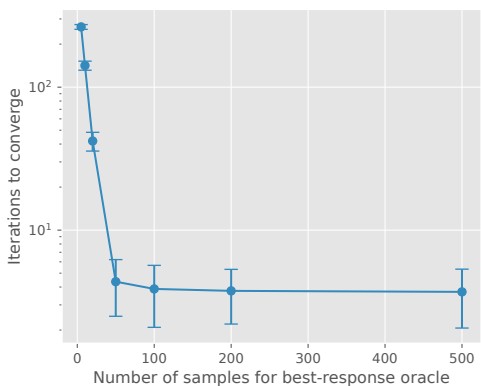

(a) The function $J$ to be optimized. Note that $J$ is not convex-concave. The saddle-point is located within the red rectangle: for any value of $\pi$ inside this region, all values of $\theta$ result in $J(\pi, \theta) = 0$, which is the lowest value of $J$.

(b) Number of iterations to converge as a function of the number of samples used for the sampling-based best-response oracle approximation. Note that the iterates never diverged during the experiments.

Figure 31: Numerical experiment on a toy example to investigate the sensitivity of the saddle-point finding iterates to violated assumptions.

## I  EXPERIMENT ON BILINEAR GAMES

We investigate whether using a finite-sample approximation of the expectation in (12) can empirically still leads to a convergence scheme on a convex-concave problem by testing it on the following bilinear game:

$$\max_{\pi} \min_{\theta} \pi \cdot \theta. \tag{92}$$

We compare gradient descent-ascent, (12) with exact expectations and (12) with a single-sample approximation of the expectation, and plot the last-iterate and average-iterate errors in $\pi$ in (32).

Note that gradient descent-ascent does not converge, as is well known in the literature (de Montbrun & Renault, 2022). Moreover, as guaranteed by Section C, the average-iterates of (12) converge. While using a single-sample approximation does slow down the convergence rate, it does seem to still converge empirically. Investigating this theoretically is out of scope for the current work.

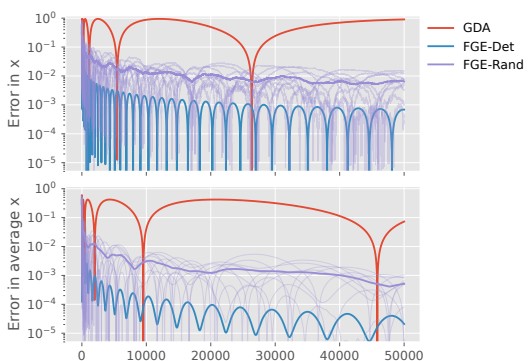

Figure 32: Last-iterate and average-iterate errors on a bilinear game.

