# OpenReview forum: "Solving Parameter-Robust Avoid Problems with Unknown Feasibility using Reinforcement Learning"
_ICLR.cc/2026/Conference — ICLR 2026 Poster_

### Official Review · Reviewer_kNnA · 2025-10-31

**Soundness:** 2
**Presentation:** 3
**Contribution:** 3
**Rating:** 6
**Confidence:** 3

**Summary:**

The paper proposes an exploration method for identifying a subset of feasible initial conditions in the parameter-robust avoid problem. The core component is a conservative feasibility classifier that incrementally expands the estimated feasible region from collected samples. The policy is updated by approximately solving a robust policy optimization problem and then executed to gather new data for further classifier refinement. Across MuJoCo and other control benchmarks, FGE achieves solid safe coverage compared to standard baselines.

**Strengths:**

- It's interesting how the paper connects feasible-set estimation with robust policy optimization. The formulation naturally follows from the parameter-robust avoid problem and builds a nice bridge between reachability analysis and safe RL.

- The proposed method is empirically validated on several benchmarks, showing consistent improvements in safe coverage. Theoretical claims are somewhat idealized but conceptually sound (see Weaknesses for details).

- Overall, the paper offers a useful approach for safe exploration that explicitly tackles feasibility estimation, even if its scalability to high-dimensional parameter spaces remains unexplored.

**Weaknesses:**

- Theorem 1’s guarantees (zero false positives and controllable false negatives) hold only for a fixed policy $\pi$. But, FGE continuously updates $\pi$ using PPO, making the rollout distribution $\rho$ and the conditional success probability non-stationary. This violates the theorem’s assumptions and undermines the claimed guarantee.

- Since FGE only samples parameters classified as infeasible, any false positive (infeasible but predicted feasible) is never revisited. This can cause the algorithm to miss meaningful parts of the parameter space and under-cover the true feasible set.

- If a single rollout happens to succeed by chance (e.g., due to disturbances or randomness), that parameter is permanently added to $D_f$. These accidental successes can cause the estimated feasible set to become larger than it actually is, especially near boundaries where the system is sensitive to small variations.

- All experiments use low-dimensional parameter spaces. Performance in higher-dimensional or stochastic settings remains untested.

**Questions:**

- How do you justify using Theorem 1 although $\pi$ and $\rho$ are continuously updated during training? Have you considered an online learning or bounded drift variant of the guarantee?

- Did you observe blind spots caused by false positives in practice? Could uncertainty-based or boundary-based focused sampling (e.g., entropy or ensemble variance) alleviate this issue?

- How sensitive is performance to occasional mislabeling from accidental successes? Have you tried multi-rollout or confidence-based labeling?

- Have you tested FGE in higher-dimensional parameter spaces (at least in 5D)? What are the computational or sample-efficiency challenges as the parameter dimension grows?

---

> ### Author Response · Authors · 2025-11-21
> **Author's Response to kNnA (1/2)**
>
> Thanks for recognizing our work for its conceptually sound theoretical claims and nice bridge between reachability analysis and safe RL! In our response, we:
> - **Run a new experiment** on a higher-dimensional (9D) parameter space
> - Discuss how the computational and sample effiency scale in the parameter dimension
> - Discuss how Theorem 1 interacts with changing $\pi$ and $\rho$
> - Discuss how mispredictions are dealt with in FGE
> - Clarify how FGE handles false-positives
> - Clarify that "accidental successes" do not occur under deterministic dynamics
>
> &nbsp;
>
> ## Details
>
> > ### **(W4, Q4):** How does FGE perform in higher-dimensional parameter spaces (at least in 5D)?
>
> This is a good idea. We introduce a new environment `Lander9D` with a **9D parameter space** (and also uses RGB images as the observation space) and **perform a new experiment** comparing our method FGE against the two best-performing (VDS, PLR) as well as DR. We show the results in **Fig. 9** in the revised pdf as well as in the table below.
>
> |Name|Safety Rate (↑)|Coverage Gain vs DR (↑)|Coverage Loss vs DR (↓)|Unique Coverage (↑)|
> |-|-:|-:|-:|-:|
> |FGE|0.94|0.79|0.04|0.47|
> |DR|0.90|0.00|0.00|0.16|
> |VDS|0.93|0.72|0.04|0.35|
> |PLR|0.62|0.12|0.33|0.01|
>
> FGE achieves both the highest safety rate and highest unique coverage, providing evidence that **FGE can scale to higher parameter spaces**.
>
> &nbsp;
>
> ---
>
> > ### **(Q4):** What are the computational or sample-efficiency challenges as the parameter dimension grows?
>
> FGE does not face any computational challenges that scale with parameter dimension.
>
> The sample-efficiency will be affected, as increasing the parameter dimension will require exponentially more draws from the base distribution to "cover" the parameter space with the same density.
> However, **this is a drawback that all methods face**. FGE can actually improve sample-efficiency, since it minimizes sampling parameters that are "easy" and well-performing under the curent policy.
>
> &nbsp;
>
> ---
>
> > ### **(W1, Q1):** How do you justify using Theorem 1 even though $\pi$ and $\rho$ are continuously updated during training?
>
> This is a valid concern, since Theorem 1 holds for a given $\pi$ and $\rho$, but these are continuously updated during training.
> Fortunately, the classifier accuracy interacts "well" with the changes in $\pi$ and $\rho$ in our setting.
>
>
> Intuitively, the main properties we rely on are the following:
> - From (14), the classifier error scales linearly in the safety rate of the policy $\pi$, i.e,.
> $$1 - q_{\psi^*}(f=1 | \theta_f) \propto 1 - p^\pi(f=1|\theta)$$
> Thus, if we fix $\rho$, **as long as the policy improves, the classifier accuracy will also improve correspondingly**.
> - Similarly from (14), the error also scales linearly with the density of $\rho$ on the feasible parameters. Thus, if we fix $\pi$, the **classifier accuracy improves if the density of $\rho$ on feasible parameters decreases**.
>     - This is **synergistic** with the _exploration distribution_ part of our parameter sampling distribution $\rho$, since it uses rejection sampling to avoid sampling from parameters currently estimated to be feasible (i.e., zero density)!
>
> Thus, assuming that the policy improves over time, these two properties give us confidence that the learned classifier will also behave well over training.
>
> &nbsp;
>
> ---
>
> > ### **(Q1):** Have you considered an online learning or bounded drift variant of Theorem 1?
>
> We have not considered this type of variant but it could be interesting. Could you clarify what you mean?

---

> > ### Author Response · Authors · 2025-11-21
> > **Author's Response to kNnA (2/2)**
> >
> > > ### **(W2, Q2):** How do false positives (infeasible parameters that are predicted feasible) impact the performance of FGE?
> >
> > Thank you for bringing up this concern! While the optimal classifier does not predict false-positive parameters, _in practice_ this can happen due to the use of neural network classifiers.
> >
> > First, we note that **FGE does sample parameters that are classified as feasible** since the final sampling distribution for $\theta$ includes the rehearsal buffer (Section 4.1) and the base distribution (e.g., see Figure 3). The resulting mixture distribution has **equal support as the original base distribution**.
> >
> > Thus, false-positive parameters can and will be sampled by FGE.
> > 1. When this happens, these samples will be unsafe since they are infeasible parameters and hence have negative labels
> > 2. These false-positive parameters will then be used for training the classifier as part of the mixture distribution $p_{\text{mix}}$ with probability $\alpha$, correcting for the classifier's error
> >
> > This feedback mechanism ensures that **false positives do not meaningfully impact the performance of FGE** and does not cause FGE to miss meaningful parts of the parameter space. This conclusion is backed empirically by our experimental results, where FGE consistently has very low Coverage Loss compared to the other baseline methods (Fig. 5b).
> >
> > We realize that our current presentation may be misleading and **have clarified this in the revised pdf in Section 4.4.**
> >
> > &nbsp;
> >
> > ---
> >
> > > ### **(W3, Q3):** How sensitive is performance to occasional mislabeling from accidental successes? Have you tried multi-rollout or confidence-based labeling?
> >
> > Great question! We focus on **deterministic dynamics**, so a rollout cannot succeed by chance on an infeasible parameter, and hence will not be added to $D_{\mathfrak{f}}$.
> > Since this assumption can be quite strict in general for RL, we discuss this in the Limitations of our work.
> >
> > While outside our current scope, an extension to the case of stochastic dynamics would require a more precise definition of what "safety" means, and would probably require some type of multi-rollout or confidence-based labeling techniques as you have mentioned.

---

> > ### Comment · Reviewer_kNnA · 2025-11-28
> > **Re: (Q1)**
> >
> > What I meant was just a version of the guarantee that works even when \pi and \rho keep changing during training. Right now Theorem 1 assumes they're fixed, but in FGE they update every iteration.  An "online" or "bounded-drift" version would basically say: as long as the policy doesn't change too much each step, the classifier's guarantee still holds over time.  I was curious if something along those lines might be possible.

---

### Official Review · Reviewer_6h75 · 2025-11-01

**Soundness:** 3
**Presentation:** 3
**Contribution:** 2
**Rating:** 6
**Confidence:** 3

**Summary:**

The paper studies a special case of a safety problem in the context of decision making, that the authors call `parameter-robust avoid` problem. The goal is to identify a subset of parameters/initial conditions, which lead to the existence of a safe policy. More specifically, a safe policy is the policy that avoids an unsafe set for *all* parameter values.

The authors first reformulate the problem as a maximisation of the subset of the parameters subject to constraints: first hitting time of the avoid set and dynamics constraints. Then the authors provide a two-stage algorithm common for robust parameter optimization problems, where they optimize over a safe policy and maximise the support of the safe parameters.

**Strengths:**

* The paper is solving a complex problem with an interesting solution, and good results
* There’s a good ablation study
* The exposition of results is great! I wish more papers would present the results this way. There’s a thesis of what algorithm does better than competitors and the supporting evidence.

**Weaknesses:**

* The paper’s flow can be improved as it is not easy to understand the problem and the contribution at the first read. While it’s not easy to write a complicated contribution in an easy way, some steps could be taken:
     * Provide a concrete example of the problem - why is it important?
     * Try to avoid a bottom up approach to explain the solution as much as possible. When there’s a lot of steps to get from point A to point B, the reader may lose the thread of these explanations. For instance,
          - I think formulation (6) can be explained intuitively without Lemmas and Theorems, which will give the reader an idea of what’s going to happen. Then explaining how we get there, will give the reader the context (if the reader is interested).
          - The same is true for Section 4.1. In this case, for example, the regularizer `\psi` is not even defined when introduced, and it’s not entirely clear how this algorithm would work. Is there iteration, stopping conditions, do we sample different `\theta` on every step?
* It would be good to discuss  the problem definition in lines 121-124 in a bit more detail. What are the assumptions on \Theta, h, f. Why is this problem hard?
* Why mention saddle-point convergence if the convergence guarantees are very restrictive? The authors mention that actually, but it all becomes confusing fast. I’d suggest having this discussion in conclusions/limitations.
* What’s the main idea of the parameter set estimation algorithm? Overall, I found this section hard to understand.
* It is not clear what the RL formulation(s) of the problem is. What formulations were used for baselines?
* I don’t quite understand what coverage is. It seems this metric depends on the algorithms, which is not easy to understand. Do we have access to the ground truth safe parameters? Is it possible to estimate the ground truth set of safe parameters?

**Questions:**

* I find it confusing how the authors define reachability. I looked in Tomlin et al 2000, and they have a more common definition of reachability: the set of states that the dynamical system can reach with a control policy. Safety is typically not a requirement for reachability definition. I would also point out that Tomlin et al 2000 study discrete-event systems and their safety definition is not directly applicable to any dynamical system. I suggest simply formally providing a definition of the authors want to study, even if these concepts are inspired by Tomlin et al 2000.
* Could you explain intuitively Section 4.2
* How is policy optimization performed? It seems it’s just a gradient descent algorithm. Did the authors use PPO, Reinforce or something custom here?
* Line 470. This is a bit confusing, I read it first as your method outperforms NSF just marginally. But this doesn’t seem to be the case! I’d recommend a more direct language here: your approach outperforms significantly both NSF and PLR.
* Small issues:
    * Lemma 1. Technically, in Lemma 1 equation 4 is not describing a sublevel set for V(x, \theta) but a level set.
    * Eq 8. `\Psi` is not defined

---

> ### Author Response · Authors · 2025-11-21
> **Author's Response to 6h75 (1/2)**
>
> Thank you for your thoughtful feedback, not only for noting that our work solves a “complex problem with an interesting solution” and praising our exposition of results and ablation study, but especially also with providing constructive criticisim on how to improve the paper's flow!
>
> Below, we address each of the issues raised in the review.
> We have updated the manuscript accordingly, and for clarity we highlight all revisions in red 🔴.
> In particular, we:
> - Improve the flow of the problem formulation and method inspired by your suggestions
> - Clarify the connection between reachability and safety in our problem setting
> - Give better explanations for feasible parameter set estimation
> - Clarify how policy optimization is performed
> - Clarify what problem formulation the baselines use
> - Explain that coverage is the safety rate, and that we have renamed this for better clarity
> - Clarify our rationale for providing saddle-point convergence guarantees
>
> &nbsp;
>
> ## Details
>
>
> > ### **(W4, Q2):** Could you provide intuition for the feasible parameter set estimation algorithm in Section 4.2?
>
> Sure!
>
> - The main challenge with feasible set estimation is that we need to solve a classification problem with only positive (feasible) labels. This is because we only know a parameter is feasible if it has been observed to be safe.
> - To solve this, we use a _mixed_ dataset:
>     - Sample data from the original dataset (with only positive labels) with probability $\alpha$
>     - Sample data from another dataset (with both positive and negative labels) with probability $1-\alpha$
> - While this may seem like a bad idea, if we choose the other dataset to be the on-policy rollout outcomes, it turns out that **this gives us a conservative classifier (no false positives) with a controllable false-negative rate that depends on $\alpha$** (Theorem 1).
>
> We agree that this section can be difficult to understand and have **reworded Section 4.2 to better convey the intuition** in the revised pdf.
>
> &nbsp;
>
> ---
>
> > ### **(Q3):** How is policy optimization performed? It seems like it is just a gradient descent. Did the authors use PPO here?
>
> Good observation!
>
> - Equation (8) is a gradient descent step
>     - (gradient ascent in the revised pdf since we changed from minimizing cost to maximizing reward) on the objective
> - We implement this as a policy update step using PPO (and have made this clear in (9) in the revised pdf)
> - Note that we can change PPO to any other on-policy algorithm in FGE
>     - See Appendix D.4 for a case study using on-policy safe RL algorithms
>
> We realize this may be confusing and have added additional clarifications to the saddle-point finding part of the revised pdf in Line 242
>
> &nbsp;
>
> ---
>
> > ### **(W5):** What formulations were used for the baselines?
>
> Good question!
>
> All baselines optimize the same RL objective defined in Equation (9) of the revised pdf, differing in the distribution over the parameters $\theta$ used.
>
> We realize this was not conveyed clearly in the paper and have **clarified this in both Section 5 and in App. G.2 in the revised pdf.**
>
> &nbsp;
>
> ---
>
> > ### **(W6):** What is coverage?
>
> Coverage is defined as the fraction of all parameters that a policy can render safe (Line 365 in the original PDF, App. G.1 in the revised pdf).
>
> - Since not all parameters $\theta$ are feasible, the highest achievable safety rate for each environment will be different depending on how large the feasible set of parameters is for that environment.
> - To enable fair comparison across environments, we instead measure the safety rate conditioned on the parameter being feasible.
> - We don't know the set of feasible parameters, so we conservatively estimate that a parameter is feasibile if any method can make it safe.
>
> To minimize confusion, **we have replaced  _total coverage_ with _safety rate_ in the revision**. We recognize that this can be unintuitive, and **have included additional explanations in Section 5 (Evaluation Metrics) of the revised pdf**.

---

> > ### Author Response · Authors · 2025-11-21
> > **Author's Response to 6h75 (2/2)**
> >
> > > ### **(Q1)**: How is the problem formulation connected to reachability in (Tomlin et al., 2000)? I suggest including a formal definition.
> >
> > Thank you for the recommendation!
> > **We have added a formal definition in Line 143 connecting reachability to our problem setting.**
> >
> > - We cite Tomlin et al. (2000) in the introduction as they were one of the first works on reachability. From their abstract:
> > > We translate **safety specifications** into **restrictions on the system’s reachable sets of states**. Then, using analysis
> > based on optimal control and game theory for automata and continuous dynamical systems, we derive Hamilton–Jacobi equations
> > whose solutions describe the **boundaries of reachable sets**.
> >
> > - Tomlin et al. (2000) focuses on hybrid systems, which **include discrete-time dynamics as a special case**
> >     - This can be done (using their notation) by having "Inv" be the empty set to disallow continuous evolution, and using $R$ to encode the discrete-time dynamics.
> >
> > &nbsp;
> >
> > ---
> >
> > > ### **(W3):** Why mention saddle-point convergence if the convergence guarantees are very restrictive?
> >
> > Great question.
> > Although the theoretical setup is idealistic, we provide it as a means to ground our method in an algorithm that is guaranteed to converge on at least some simple set of problem classes.
> >
> > Note that the assumptions for the saddle-point convergence are **sufficient but not necessary**:
> > - Violating the assumptions does not mean the algorithm does fails to converge in practice
> > - We performed an **additional experiment** on a toy example to see what happens when the assumptions of convex-concavity and access to exact best-response oracles are violated in Appendix H.
> > - Surprisingly, the saddle-point finding always converged empirically even though the assumptions are violated.
> >
> >
> > &nbsp;
> >
> > ---
> >
> >
> > > ### **(W1-2, Q4, Q5):** The paper’s flow can be improved as it is not easy to understand the problem and the contribution at the first read.
> >
> > Thank you for your suggestions, they are very helpful! We have improved the paper's flow in the revision, including addressing the following:
> > - Provide a concrete example of our problem and why it is important/hard in Line 146
> > - Included more intuitive explanations to Section 4.1, and a concrete numerical example of how the FTRL algorithm can work in App. H
> > - Updated the wording in the NSF ablation of Section 5.2
> > - Updated Line 175 to be level set instead of sublevel set
> > - Defined $\psi$ in Equation(8)

---

> ### Comment · Reviewer_6h75 · 2025-11-22
> **a few comments**
>
> Thank you for the rebuttal and the effort put into this work
>
> I'll start with the following remarks.
>
> **Re reachability**
>
> Eq 2 defines reach-avoid problem: finding the reachable set of policies under the condition of avoiding a set. It's not pure reachability problem, but close enough. Interestingly, the authors point to line 143 for the formal definition of reachability in their rebuttal. However, their definition of reachability doesn't use the word "reachability" but defines *parameter-robust avoid* problem.
>
> According to Tomlin et al 2000  (page 15 in [top of Chapter 4](https://people.ece.ubc.ca/moishi/eece571m/articles/Tomlin00.pdf))
> > A state of a dynamical system is defined to be reachable if there is an execution of the system which touches it.
>
> They use the terms `execution`, `touches it` since they are considering  `discrete-event systems with continuous nonlinear dynamics` (or hybrid systems as the authors mentioned). Tomlin et al weren't one of the first to study reachability in control theory, but their approach to `safe`  reachability of discrete-event or hybrid systems was novel.
>
> In control theory in general, reachability refers to all the states that are reachable from the initial condition under a feasible control law. It was studied in for example [1] - 30 years prior to Tomlin et al. It is not necessary to cite [1] of course - just an example.
>
> Taking all of this into account I insist correcting all the incorrect statements. For example
>
> * Line 14 *"reachability seeks to maximize the set of states from which a system remains safe indefinitely"* is incorrect under common definition of reachability in control theory (which Tomlin et al also employs)
> * Line 36 *"Hamilton-Jacobi (HJ) reachability analysis is a powerful tool for obtaining control policies and reasoning about the largest set of initial conditions from which a dynamical system can satisfy safety constraints for all time (Tomlin et al., 2000)"* is incorrect. While reachability analysis can be used for this goal as Tomlin et al showed, control policy design and safety *is not a necessary part* of reachability analysis.
> * `Line 68 *"This perspective aligns directly with the original reachability goal of worst-case safety"* reachability analysis can involve safety by considering avoid sets, but it's not the goal of reachability, in general.
>
> There are more statements like this.
>
> While this remarks may seem not as important, I would have to insist on correct statements and correct attributions of the definitions.
>
> [1] Bertsekas, Dimitri P., and Ian B. Rhodes. "On the minimax reachability of target sets and target tubes." Automatica 7.2 (1971): 233-247.
>
> **Re saddle point**
>
> I think mentioning the saddle-point convergence and convex-concavity is misleading. Convex-concavity is important for gradient descent-ascent methods, but you don't use this approach. In fact, I am not sure if you algorithm converges in convex-concave problems in general. I think it would depend on the batch size and sampling distribution for `\theta`. In practice, if you uniformly sample one `\theta` at every step in a convex-concave problem, I don't see how it will converge in general. I am sure it's not hard to construct an example like this.
>
> Again I suggest not making this a point of contention but simply remove the statements without supporting evidence.
>
> I think empirical evidence would be sufficient in this case.
>
> **Re safety rate**
> > Safety Rate This is the normal safety rate, conditioned on if a parameter is estimated to be feasible
>
> Please define what does it mean "estimated". For example, as you mentioned above: if a parameter was shown to be feasible by a policy.

---

> > ### Author Response · Authors · 2025-11-24
> > **Author's Response to [a few comments] (1/2)**
> >
> > Thank you for the fast response! We appreciate and stand behind your insistence on making correct statements and attributions.
> >
> > &nbsp;
> >
> > ---
> >
> > ### (**Re reachability**)
> >
> > We agree that our current usage of the term reachability could have led to unintentional confusion. There are works that treat safety and controller synthesis as "part" of reachability. For example, in [A]
> >
> > > Hamilton-Jacobi (HJ) reachability analysis is a verification method for guaranteeing performance and **safety properties** of systems
> >
> > > Thus, reachability analysis gives us the **safe set** as well as a **controller** that will keep the system in the safe set
> >
> > in [B]:
> >
> > > Hamilton-Jacobi (HJ) reachability analysis is a rigorous tool that verifies the **safety** and/or liveness of a dynamic system
> >
> > in [C]:
> >
> > > Hamilton-Jacobi (HJ) Reachability analysis is a powerful and effective approach for analyzing and **controlling** hybrid dynamical systems.
> >
> > &nbsp;
> >
> > On the other hand, as you have mentioned, in other works, reachability only refers to computing (potentially approximations of) forward/backward reachable sets, e.g., as you have cited in (Tomlin et al., 2000) [D]
> >
> > > A state of a dynamical system is defined to be reachable if there is an execution of the system which touches it.
> >
> > or for example in Matthias Althoff's thesis [E]
> >
> > > Reachability analysis consists in computing the reachable set of a dynamical system.
> >
> > where reachability can be used for safety / controller design, but these are not part of the _definition_ of reachability.
> >
> > We acknowledge that the control community **does not have a unified consensus on whether safety / controller synthesis falls under the definition reachability or not.**
> >
> > To prevent further confusion, we disambiguate this in our manuscript by **changing instances of "reachability" or "HJ reachability" to instead refer to $\color{#913AA0}{\textsf{optimal safe controller synthesis}}$** and have denoted these changes in $\color{#913AA0}{\textsf{purple}}$.
> > For example, with the sentences you have brought up:
> >
> > - **Before:** _reachability_ seeks to maximize the set of states from which a system remains safe indefinitely
> >     - **After:** $\color{#913AA0}{\textsf{optimal safe controller synthesis}}$ seeks to maximize the set of states from which a system remains safe indefinitely
> >
> > - **Before:** _Hamilton-Jacobi (HJ) reachability analysis_ is a powerful tool for obtaining control policies and reasoning about the largest set of initial conditions from which a dynamical system can satisfy safety constraints for all time
> >     - **After:** $\color{#913AA0}{\textsf{Hamilton-Jacobi (HJ)-based optimal control analysis}}$ is a powerful tool for obtaining control policies and reasoning about the largest set of initial conditions from which a dynamical system can satisfy safety constraints for all time
> >
> > - **Before:** This perspective aligns directly with the original _reachability_ goal of worst-case safety
> >     - **After:** This perspective aligns directly with the original goal of worst-case safety
> >
> > [A] Bansal, Somil, et al. "Hamilton-jacobi reachability: A brief overview and recent advances." 2017 IEEE 56th Annual Conference on Decision and Control (CDC). IEEE, 2017.
> >
> > [B] Ganai, Milan, Sicun Gao, and Sylvia L. Herbert. "Hamilton-jacobi reachability in reinforcement learning: A survey." IEEE Open Journal of Control Systems 3 (2024): 310-324.
> >
> > [C] Javier Borquez, Shuang Peng, Yiyu Chen, Quan Nguyen, and Somil Bansal. Hamilton-Jacobi Reachability Analysis for Hybrid Systems with Controlled and Forced Transitions. In Proceedings of Robotics: Science and Systems, Delft, Netherlands, July 2024
> >
> > [D] Tomlin, Claire J., John Lygeros, and S. Shankar Sastry. "A game theoretic approach to controller design for hybrid systems." Proceedings of the IEEE 88.7 (2000): 949-970.
> >
> > [E] Althoff, Matthias, Goran Frehse, and Antoine Girard. "Set propagation techniques for reachability analysis." Annual Review of Control, Robotics, and Autonomous Systems 4.1 (2021): 369-395.

---

> > > ### Author Response · Authors · 2025-11-24
> > > **Author's Response to [a few comments] (2/2)**
> > >
> > > ### (**Re saddle point**)
> > >
> > > We agree that statements should not be made without supporting evidence and **would be happy to remove or reword any such statements**. **Could you clarify exactly which claims we make is not supported by evidence?**
> > >
> > > 1. The iterates (6) and (7) **have average-iterate convergence** by using existing results from (Orabana, 2019), which we have shown in Appendix C.
> > > 2. Equation (8), obtained by replacing (7) with an expectation (if we evaluate expectation exactly), **still retains the convergence results** since it is equivalent to (7).
> > > 3. When we use a Monte Carlo approximation of the expectation in (8), then it is no longer equivalent to (7).
> > >     - However, we can use techniques from [A] to show that the resulting regret minimizer still satisfies sublinear regret _with high probability_. This will still allow us to use Lemma 2 to show, _with high probability_, that the average iterates converge to a saddle point.
> > >
> > > The convex-concavity assumption comes from (Orabona, 2019) and is needed to prove Lemma 2 (i.e., we can bound the duality gap with the regret of the $\mu$ player) since the proof invokes Jensen's inequality.
> > >
> > > Of course, when we make further approximations (approximating the best-response oracle for $\theta$, using the policy update step of PPO instead of exact gradient descent for FTRL), then proving convergence under these conditions is not feasible. However, the theoretical results still provide insight on why our method works well empirically.
> > > **This is already made explicit in both the Section 4.1 and in the limitations section**.
> > >
> > >
> > > &nbsp;
> > >
> > > As a sanity check, we perform a simple empirical test for whether (8) with exact expectations and with a one-sample estimate converges on convex-concave games. We consider the following bilinear example:
> > >
> > > \begin{equation}
> > > \max_{\pi \in [0,1]} \min_{\theta \in [0,1]}\\; \pi \cdot \theta
> > > \end{equation}
> > >
> > > The objective here is convex-concave and is Lipschitz and thus satisfies the assumptions required for convergence in Appendix C, with an optimal solution at the origin $(\pi, \theta) = (0, 0)$.
> > >
> > > We compare:
> > > - gradient descent-ascent
> > > - Equation (8) using exact expectations
> > > - Equation (8) using a single-sample estimate
> > >
> > > and plot the results in Appendix I.
> > >
> > > As **guaranteed by Appendix C**, the average-iterates of (8) converge. While using a single-sample approximation does slow down the convergence rate, it does still converge, supporting the theoretical arguments.
> > >
> > > [A] Farina, Gabriele, Christian Kroer, and Tuomas Sandholm. "Stochastic regret minimization in extensive-form games." International Conference on Machine Learning. PMLR, 2020.
> > >
> > >
> > > &nbsp;
> > >
> > > ---
> > >
> > > > ### (**Re safety rate**) Please define what does it mean "estimated". For example, as you mentioned above: if a parameter was shown to be feasible by a policy.
> > >
> > > This is **already** given in both the main text (Line 408 in the revised pdf):
> > >
> > > > we conservatively estimate that a parameter is feasible if any method can make it safe.
> > >
> > > and also in the text surrounding equation (58) in the Appendix G.1:
> > >
> > > > let $\mathfrak{s}_{\text{any}}$ denote the binary random variable that is 1 if, given a random $\theta$, any method (including
> > > all ablations), over any seed, can keep $\theta$ safe.
> > > >
> > > > ...
> > > >
> > > > This is the normal safety rate, conditioned on if a parameter is estimated to be feasible. $$\text{SafetyRate}_m = p(\mathfrak{s}_m=1 | \mathfrak{s}\_{\text{any}}=1)$$

---

> ### Comment · Reviewer_6h75 · 2025-11-24
>
> **Re reachability**
> > We acknowledge that the control community does not have a unified consensus on whether safety / controller synthesis falls under the definition reachability or not.
>
> I don't agree with this statement. One can have reachability based control synthesis, but reachability analysis, in general, doesn't necessarily include safety/controller synthesis.
>
> I am fine with the authors revisions. However, if they authors want to use the word "reachability" in the abstract or introduction, I am not against it. My main complaint was a sweeping statement over reachability without qualification in the main text. I will leave this decision on using reachability to the authors. I think I convinced the authors of this small ambiguity in notation and trust them to make the changes appropriately.
>
> **Re saddle point**
>
> Apologies for misunderstanding their statements, all the statements are supported by evidence in fact. I suggest rephrasing the main text to reflect that the convergence result in Appendix is valid for minimization through sampling.
>
> **Re  safety rate**
>
> Thank you for expanding the definition. Just to expand on the reason for this request: when a reader misses some details, they would look for a definition to provide all the information needed for the term (similarly, the captions to Figures should be self contained). To avoid losing such a reader I would always suggest adding all the needed for a definition. Or at least close by. You would avoid many requests for clarification during the review as well.

---

> > ### Author Response · Authors · 2025-11-24
> >
> > Thank you again for your fast response! We are happy our clarifications were able to answer your questions.
> >
> > &nbsp;
> >
> > (**Re saddle point**) Per your suggestions, we have clarified the main text to reflect that the convergence results in Appendix C are valid even under Monte Carlo approximation of the expectation in (8):
> > - Line 240: "Using techniques from (Farina et al., 2020), a Monte-Carlo approximation of the expectation in (8) still results in sublinear regret with high probability for $\pi$ and hence still converges to the saddle-point with high probability."
> >
> > &nbsp;
> > ---
> >
> > Since your current score is a 6 (_marginally above the acceptance threshold_), **what final concerns are holding you back from further raising the score?** We would be more than happy to perform additional experiments and further clarify aspects in the paper to address them.

---

> > > ### Comment · Reviewer_6h75 · 2025-11-26
> > >
> > > Thank you for your responses!
> > >
> > > I will maintain my score at the moment. For what it's worth I think the idea is great and the results are good as well!
> > >
> > > For a more solid score, the flow of the paper would have be improved. Like I mentioned bottom-ups approach are harder to understand and write in most of the cases. Not sure if it can be solved during peer-review period however. I do like the presentation of results, however, but it's not enough to bring it over the line, unfortunately.

---

### Official Review · Reviewer_CPF8 · 2025-11-06

**Soundness:** 2
**Presentation:** 2
**Contribution:** 2
**Rating:** 4
**Confidence:** 4

**Summary:**

This work proposes a method (called FGE) for solving an initial-states-robust avoid (reachability) problem when feasibility of the initial states set is unknown. To achieve this, it jointly learns (a) a classifier that identifies which parameters (defining initial states) admit a safety-preserving policy and (b) trains a robust policy over that discovered feasible subset by alternating approximate saddle-point / best-response updates and a replay/rehearsal scheme. FGE adapts the reset (initial-state) distribution as a mixture of a base distribution, an explore distribution (via rejection sampling against the feasibility classifier) to expand the feasible set, and a rehearsal buffer to stabilize robust optimisation. The paper then provides theoretical guarantees on the classifier and empirical evidence across several MuJoCo and vehicle/aircraft tasks that FGE yields substantially larger safe (feasible) initial state sets than baselines.

**Strengths:**

- **Theoretical grounding:** The paper gives a clear problem formulation, shows equivalence between its indicator-style objective and a reachability formulation (Lemma 1), and analyzes properties of the learned feasibility classifier (Theorem 1), which underpin the empirical design choices
- **Clear, practical algorithmic pipeline:** FGE is presented as an algorithm that can be dropped on top of any on-policy method (PPO in the experiments), combining feasibility learning, rejection-sampling exploration, and a rehearsal buffer. This is an appealing and implementable recipe. Algorithmic pseudocode and reset-distribution mechanics are provided.
- **Several empirical evaluations and useful metrics:** Experiments cover multiple domains (ToyLevels, Dubins, Hopper, HalfCheetah, FixedWing) and a broad set of baselines. The coverage metrics and derived diagnostics (Coverage Gain vs DR, Coverage Loss vs DR, etc) are insightful, especially the Coverage Gain vs DR plots that highlight FGE’s ability to find rare hard initial states. Several ablations (e.g. explore / rehearsal / classifier choices) are also provided and are informative.
- **Significance for safe RL:** The paper tackles an important, well-motivated gap between reachability/robust objectives and the sampling of initial states in RL: with unknown feasibility of those initial states. The proposed formalization may be a useful addition to the safe-RL literature.

**Weaknesses:**

- **Notation and readability:**
  - The abstract and introduction are hard to read because of vague sentences. For example, there is repeated use of terms like "initial conditions" and "initial parameters" throughout without defining what they mean in the context of this paper (this is only done in the next section). I initially thought "initial conditions" meant the initial conditions of an optimisation process, and not the distribution/set of initial states. Similarly, I thought initial parameters meant the initial parameters of the policy neural network, but instead it meant some parameters that define the deterministic dynamics function and specific initial state for an episode.
  - The problem formulation was also extremely confusing. The paper uses notations from control theory instead of RL (despite the RL focus), and uses the usual RL notations for something else. For example, A and S are not action and state spaces but instead avoid states and safe initial states respectively. Also the value function is not the RL value function (it is not an expectation of cumulative rewards and does not necessarily satisfy the Bellman equations). This just makes readability extremely more difficult than it has to be as the paper progresses. It would have been much clearer if the authors just used standard RL notations and terminology. For example the problem setting seems to be essentially a deterministic contextual MDP with an almost-surely safe RL objective (the agent needs to avoid unsafe states with probability 1).
  - Minor: Lots of typos and mathematical impreciseness. E.g.
    - Inconsistent use of = and := (both are being used for definitions). The general convention is using = to claim equality and := to define a function.
    - Equation 1 $f_{\theta}$ and $x_0$ are undefined (explicitly say what they are).
    - Lines 130 and 144 $V(x_0;\theta)$ should be $V(x;\theta)$.
    - $T_{\theta}$ should depend on $x$.
    - Line 1348 "that that".

- **Scope limitations (method assumptions):**
  - The approach is only applicable to episodic deterministic environments where the agent have access to the reset function (can chose the initial state of each episode).
  - The paper defines the dynamics and policy as being conditioned on the parameter $\theta$ (which is fixed thoughout an episode), but the experiments appear to only change the start sates with $\theta$. It is unclear what happens when the environment dynamics also change with $\theta$, as in most contextual MDPs.
  - It is also unclear how FGE scales to high-dimensional observations such as RGB pixels (the experiments seem to use state vectors). The feasibility classifier and rejection sampling may become challenging in very high-dim $\theta$ or when observations are high-dimensional.

- **Theory–experiment alignment:** The theoretical setup conditions (convexity/concavity, access to best-response oracles, deterministic dynamics) needed for convergence claims are not satisfied in the empirical setups; while the paper acknowledges this, it should more explicitly describe the gap and, if possible, include analyses showing sensitivity to the violated assumptions (some ablations exist but more explicit discussion is needed).

- **Experimental reporting gaps (Major):**
  - The paper does not show standard reward / success/failure training curves in the main results (only coverage metrics are emphasized). For a safe-RL paper, plots showing rewards and success/failure rates (or explicit returns vs constraint violations) would help assess whether policies genuinely optimize task objectives while avoiding unsafe states. A natural concern regarding the proposed method is that the oversampling of hard initial states may compromise policy learning for the actual task.
  - The paper also does not clearly report the reward functions used in each environment (how goal reward and avoid/safety labels are combined). Given the task descriptions, it seems like it is not just something as simple as the negative of the indicator function in Equation 3. Because the RL objective and evaluation hinge on both avoiding unsafe regions and achieving task goals, omission of explicit reward definitions (and corresponding training plots) makes it hard to judge whether FGE is improving safety, task performance, or both (which is their main motivation/claim for improving the initial states distribution).

- **Baselines and comparisons (Major):**
  - The choice of baselines leans heavily on robust RL, curriculum, and UED methods (instantiated with PPO). However, no standard safe-RL baselines are used nor cited (such as the ones in the OmniSafe benchmark, like PPO-lagrangian and CPO). More importantly, it is also unclear why the authors did not compare against works that are also focused on maximally safe policies (the setting of this paper), like Saute-RL and ROSARL.
  - This is important because (a) FGE explicitly uses the safety label / safety function to train its feasibility classifier and to guide resets  (information that many baselines may not use) and (b) the paper's stated goal is to produce safe policies. The absence of safe-RL baselines and citations to directly related safe-RL literature weakens the claims substantially.

**Questions:**

Please see the weaknesses above. Mainly:

1. How is the reward function combined with the safety constraints in each environment? If not, how do the current baselines "see" the safety constraints?
2. Could the authors provide reward and/or success-rate plots to show how the improved initial states distributions affect the safety constraints and return maximization (which is the main motivation for this work)?
3. Why were safe RL baselines (e.g., PPO-Lagrangian, CPO, Sauté-RL, ROSARL) omitted from the comparison? Would including them alter the conclusions?
4. How does FGE scale to high-dimensional observation spaces (e.g., RGB pixel inputs)?
5. In the theoretical formulation, both dynamics and policy depend on $\theta$, but the experiments vary only the start distribution. What happens when environment dynamics change with $\theta$?
6. Please fix minor presentation issues throughout (e.g., undefined variables in Eq. 1 and incorrect state variables in definitions).

---

> ### Author Response · Authors · 2025-11-21
> **Author's Response to CPF8 (1/2)**
>
> Thank you for your thoughtful feedback, especially for noting that our work “may be a useful addition to the safe-RL literature” and for highlighting its theoretical grounding, empirical evaluations, and informative ablation studies!
>
> Below, we address each of the issues raised in the review. We have updated the manuscript accordingly, and for clarity we highlight all revisions in red 🔴.
> Most notably, in addition to improving the clarity of many aspects that were unclear, we:
> - Conduct **additional experiments** on environments with **RGB image-based observation spaces**.
> - Provide an example demonstrating how **FGE can be integrated into on-policy safe-RL methods** to improve safety performance.
> - Clarify the paper to more explicitly describe the theoretical gap
> - Perform a **new numerical study** that examines the sensitivity to the violated assumptions needed for theoretical convergence
> - Add additional citations to safe RL works
> - Clarify the reward function definition
> - Clarify that the training curves are already in the paper
> - Clarify why it is sufficient to only vary the start distribution
>
> &nbsp;
>
> ## Details
>
> > ### **(W2.3, Q4):** How does FGE scale to high-dimensional observation spaces (e.g., RGB pixel inputs) and parameter spaces?
>
> A good question! We introduce a new environment `Lander2D` that uses RGB images as the observation space and **perform a new experiment** comparing our method FGE against the two best-performing (VDS, PLR) as well as DR. We show the results in **Fig. 9** as well as in the table below.
>
> |Name|Safety Rate (↑)|Coverage Gain vs DR (↑)|Coverage Loss vs DR (↓)|Unique Coverage (↑)|
> |-|-:|-:|-:|-:|
> |FGE|0.99|0.95|0.01|0.77|
> |DR|0.96|0.00|0.00|0.09|
> |VDS|0.93|0.69|0.06|0.04|
> |PLR|0.93|0.82|0.07|0.10|
>
> **FGE achieves the highest safety rate**, providing evidence that **FGE scales well to high-dimensional observation spaces such as RGB images**.
>
> We also test on a variant of this environment but with a 9D parameter space (`Lander9D`) that still uses RGB images as observations, to see how methods perform with higher dimensional parameter spaces. The results are shown in the table below:
>
> |Name|Safety Rate (↑)|Coverage Gain vs DR (↑)|Coverage Loss vs DR (↓)|Unique Coverage (↑)|
> |-|-:|-:|-:|-:|
> |FGE|0.94|0.79|0.04|0.47|
> |DR|0.90|0.00|0.00|0.16|
> |VDS|0.93|0.72|0.04|0.35|
> |PLR|0.62|0.12|0.33|0.01|
>
> The same trends hold here with **FGE achieving the highest safety rate**, which shows that FGE can also scale to higher parameter spaces.
>
> &nbsp;
>
> ---
>
> > ### **(W3):** The theoretical setup conditions (convexity/concavity, access to best-response oracles, deterministic dynamics) needed for convergence claims are not satisfied in the empirical setup. While the paper acknowledges this, it should more explicitly describe the gap
>
> Indeed, though the theoretical setup is idealistic, it is meant to ground our method in an algorithm that is guaranteed to converge on at least some simple set of problem classes.
>
> While we do not satisfy the convex-concave requirements or have exact best-response oracles, **our experiments do use deterministic dynamics**, so there is no gap there.
>
> **We have revised our pdf to explicitly describe this gap in Section 4.1**.
>
> &nbsp;
>
> ---
>
> > ### **(W3):** The paper should also include analyses showing sensitivity to the violated assumptions.
>
> Performing a theoretical analysis to show the sensitivity to violating convexity/concavity and exact oracle assumptions is out of the scope of this present work. Instead, **we perform a simple numerical study** to see what happens when we only have approximate best-response oracles.
>
> We consider the following minimax problem:
>
> $$
> \min_{\pi \in [-1, 1]} \max_{\theta \in [-1, 1]} J(\pi, \theta),
> $$
> where the function $J : \mathbb{R} \to \mathbb{R} \to \mathbb{R}$ is defined by
> $$
> \begin{aligned}
>   J(\pi, \theta) &= \mathbb{1}( h(\pi, \theta) > 0 ), \\\\
>   h(\pi, \theta) &= -1296 \pi^4 + 36 \pi^2 - 216 \pi^2 \theta + 2592 \pi^3 \theta - 4
> \end{aligned}
> $$
>
> Note that $J$ is not convex-concave. Moreover, we violate the best-response exact oracle assumptions by performing sampling-based optimization, where we vary the number of samples used to adjust the approximation accuracy. The results are shown in **App. H**.
>
> While our toy example is by no means comprehensive or representative, we find that the saddle-point finding algorithm still converges despite violating the assumptions of convex-concavity and exact best-response oracles. While larger approximation errors in the best-response oracle do result in longer convergence times, this is a graceful degradation and no divergence is observed (Fig. 29).

---

> > ### Author Response · Authors · 2025-11-21
> > **Author's Response to CPF8 (2/2)**
> >
> > > ### **(W5, Q3):** Why not compare against safe RL baselines (e.g., PPO-Lagrangian, CPO, Sauté-RL, ROSARL)?
> >
> > Your concern is valid, as our problem setting is similar to those considered by standard safe-RL methods!
> >
> > We emphasize that **FGE is orthogonal to safe-RL methods** since it only **defines the initial state distribution**.
> > As you have correctly noted in your strengths, we present FGE as "an algorithm that can be dropped on top of any on-policy method (PPO in the experiments)",
> > and **we can apply FGE to any other on-policy method including on-policy safe RL methods!**
> > Therefore, **the conclusions we draw in the paper are unaltered when also considering safe RL**.
> >
> > To illustrate this, we **perform a new experiment** where we apply FGE to PPO-Lagrangian (PPO-L) and SauteRL and examine how it performs compared to domain-randomization (DR). As both algorithms have a hyperparameter (Lagrange multiplier for PPO-L, unsafe penalty for SauteRL) that greatly affects each method's performance, we compare the _pareto frontier_ obtained by sweeping across this hyperparameter.
> >
> > We show the results in  **App. D.4** in the revised pdf, where **applying FGE to PPO-L and SauteRL greatly expands the pareto frontier compared to DR**. This shows that FGE can be used to complement existing safe RL methods to improve safety over a wider set of parameters.
> >
> > &nbsp;
> >
> > ---
> >
> > > ### **(W5):** Safe RL works are not sufficiently cited
> >
> > Thank you for the suggestion! We have **added Safe RL to the related works section** that cites the Safe RL works you have mentioned (PPO-Lagrangian, CPO, Sauté-RL, ROSARL), as well as additional works from the OmniSafe benchmark and more.
> >
> > &nbsp;
> >
> > ---
> >
> > > ### **(W4.2, Q1):** What is the reward function? Is the reward function separate from the safety constraints, as in Safe RL?
> >
> > **The reward function is the safety constraint** and **is the negative of the indicator function** in (3). We do not have a separate task performance objective.
> >
> > In our problem, we are only concerned with safety (e.g., (6) only has safety). Since there is no task performance metric, we optimize for safety as the sole objective and use it as the reward (3).
> >
> > We realize this may be unclear, and have **clarified this in the updated pdf in Section 4.1**.
> >
> > &nbsp;
> >
> > ---
> >
> > > ### **(W4.1, Q2):** Where are the success rate plots and training curves?
> >
> > These were **already included** in the original submission:
> > - The **safety rate is the same as the coverage**, and we have plotted the IQM over all environments (Fig 5b in the original and the revised pdf) as well as for each individual environment (Fig 15 of the original pdf, Fig 16 of the revised).
> > - The training curves are in Fig 16 of the original pdf, Fig 17 of the revised pdf
> >
> > To improve clarity, **we have changed all uses of the term "total coverage" to "safety rate" in the revised pdf.**
> >
> > &nbsp;
> >
> > ---
> >
> > > ### **(W2.2, Q5):** Why do the experiments only vary the start distribution, even though the dynamics can also depends on $\theta$ in the problem formulation?
> >
> > Good observation! Without loss of generality, it is sufficient to only consider initial **augmented states**, where we augment the state with $\theta$ (Line 127 in the original pdf, Line 148 in the revised pdf).
> >
> > The environment dynamics in our experiments **already change with $\theta$**:
> > - In `Dubins`, the parameters control the speed of the two uncontrolled cars.
> > - In `Cheetah`, the parameter controls the amount of force applied to the moving platform.
> >
> > We realize this may be confusing and have **clarified this in Line 166 of the revised pdf.**
> >
> > &nbsp;
> >
> > ---
> >
> > > ### **(W1, Q6):** Minor presentation issues
> >
> > Thank you for raising this concern, we apologize for the confusion!
> > We have added more intuitive explanations and fixes to the revision, including these changes:
> > - Clarify "initial conditions" and "initial parameters"
> > - Convert control notation to RL terminology throughout the paper
> > - Draw the connection between our problem setting and contextual MDPs with an almost-surely safe RL (page 3, footnote 1)
> > - Clarify the non-RL value function (page 3, footnote 2)
> > - Fix the mentioned typos and mathematical impreciseness
> >
> > Please let us know if we missed anything!

---

### Author Response · Authors · 2025-11-21
**Author response to all reviewers**

$\def\a{\color{#648FFF}{\textsf{CPF8}}} \def\b{\color{#E69F00}{\textsf{6h75}}} \def\c{\color{#DC267F}{\textsf{kNnA}}}$
$\def\feas{\mathfrak{f}}$

We thank the reviewers for their valuable comments! We are excited that the reviewers identify that we solve an interesting, useful and well-motivated problem ($\a$, $\c$), recognize that our proposed method is grounded by idealistic but sound theoretical results ($\a$, $\c$), appreciate our strong and insightful empirical results (**all** reviewers) and good presentation ($\b$, $\c$), and acknowledge our informative ablation studies ($\a$, $\b$). We believe that **FGE takes a significant step in tackling the problem of feasibility that has been underlooked in the safe RL community**.

---


As _all_ reviewers have recognized our technical novelty, the primary criticisms (raised by $\a$ and $\c$) stemmed from insufficient clarification on the presentation of our problem formulation and FGE algorithm and questions about scalability to high-dimensional parameter and observation spaces. We agree that these aspects might be unclearly stated in the initial submission, which led to unnecessary confusion and could mislead readers. This was never our intention.

In the revision, we carefully revise our abstract, introduction and problem formulation and include additional clarifications throughout to improve readability and clarity. Notable changes are marked in red 🔴 in the revision and enumerated below.

1. We now use the usual RL notations (e.g., $s$ for state and $a$ for action) in the problem formulation to improve readability ($\a$).
2. Improved the clarity of the exposition of FGE ($\b$, $\c$)
3. Added citations to safe RL works by adding safe RL to the related works ($\a$).
4. More clearly specified RL formulation of our problem and baselines ($\a$, $\b$)
5. Clarified the "coverage" metric by renaming it to "safety rate" ($\a$, $\b$)

&nbsp;

---

# New Experiments

Additionally, we perform new experiments to address concerns about scalability to high-dimensional observation spaces ($\a$), higher-dimensional parameter spaces ($\a$, $\c$), and comparisons to safe RL baselines ($\a$).

&nbsp;

## RGB pixel inputs ($\a$)

We introduce a new environment `Lander2D` that uses RGB images as the observation space and compare our method FGE against the two best-performing (VDS, PLR) as well as DR. We show the results in **Fig. 9 of the revised pdf** as well in the table below.

|Name|Safety Rate(↑)|Coverage Gain vs DR(↑)|Coverage Loss vs DR(↓)|Unique Coverage (↑)|
|-|-:|-:|-:|-:|
|FGE|0.99|0.90|0.01|0.69|
|DR|0.97|0.00|0.03|0.10|
|VDS|0.93|0.26|0.06|0.01|
|PLR|0.95|0.55|0.04|0.20|

&nbsp;

## Higher-dimensional Parameter Spaces ($\a, \c$)

We consider a `LunarHard9D` variant of the `Lander2D` which has a **9D parameter space** (and still uses RGB images as the observation space) and **perform a new experiment** comparing our method FGE against the two best-performing (VDS, PLR) as well as DR. We show the results in **Fig. 9** in the revised pdf as well as in the table below.

|Name|Safety Rate (↑)|Coverage Gain vs DR (↑)|Coverage Loss vs DR (↓)|Unique Coverage (↑)|
|-|-:|-:|-:|-:|
|FGE|0.94|0.79|0.04|0.47|
|DR|0.90|0.00|0.00|0.16|
|VDS|0.93|0.72|0.04|0.35|
|PLR|0.62|0.12|0.33|0.01|

&nbsp;

## Safe RL Comparison ($\a$)

We show how FGE can be paired with Safe-RL methods to _additionally optimize for a separate task objective_ while satisfying safety constraints by demonstrating using two safe RL methods on an augmented `ToyLevels` environment, and show results in **Appendix D.4 of the revised pdf**.
In general, our method is orthogonal to any on-policy method, including safe RL, and can easily complement them to improve safety.

---

### Author Response · Authors · 2025-12-01

$\def\a{\color{#648FFF}{\textsf{CPF8}}} \def\b{\color{#E69F00}{\textsf{6h75}}} \def\c{\color{#DC267F}{\textsf{kNnA}}}$

Of the 3 reviewers, two ($\b,\c$) acknowledged our revision and have one final concern. The remaining reviewer ($\a$) has a **misunderstanding** of our work and many of their concerns stem from this.
We have addressed all concerns in the revision and **summarize the consensus reached during discussion** and **key additional results** we have added in the revision.

&nbsp;

# Consensus Reached during Discussion Phase

## [Reviewer $\b$: Reachability, Guarantees, Paper Flow]
After our initial reply, $\b$ raised concerns which we were able to come to **consensus** on:
- **Use of the word reachability:**
    - Quote (Nov 24): "_I am **fine with the authors revisions**_"
- **Saddle point guarantees:**
    - Quote (Nov 24): "_**Apologies for misunderstanding** their statements, all the **statements are supported by evidence** in fact._"
- **Safety rate metric**
    - Quote (Nov 24): "_Thank you for expanding the definition._"

&nbsp;

$\b$ supports our work, and **would be willing to increase their score if we addressed their concerns on the paper's flow**
- Quote (Nov 26): "_For what it's worth **I think the idea is great and the results are good as well!** ... **For a more solid score, the flow of the paper would have be improved**_"

- We **address this in our latest revision** by
    - Introducing a concrete example to better illustrate the problem and use this running example to illustrate the method in the new Figure 2 and Figure 5
    - Add exposition to the start of Sections 4.1, 4.2 and 4.3 that better explains the main idea to guide the readers

&nbsp;

## [Reviewer $\c$: Scalability, False Positives, Guarantees]
Reviewer $\c$ initially raised questions on how FGE scales to higher dimensional parameter spaces and handles false-positives, which our initial reply addressed. We came to consensus on these questions, and Reviewer $\c$ **raised no further concerns on this**.

Their remaining request is for a version of the theoretical guarantees that holds even when $\pi$ and $\rho$ change during training.
- We address this in our latest revision in **Proposition 3**, where we provide conservative guarantees on classifier accuracy **even under a worse-case policy** when we use the $\rho$ that is used in practice.

&nbsp;

## [Reviewer $\a$: Scalability, Relationship to Safe RL]
We appreciate Reviewer $\a$ for raising important questions on scalability to pixel observations and higher dimensional parameters spaces, and the relationship to Safe RL.
- Many of Reviewer $\a$'s concerns arise from a **misunderstanding**:
    - Our work solves **single-objective safety problems**, but their questions ask about **multi-objective problems** with both **objectives** and **safety**
    - Nowhere in our work do we mention a separate "task objective"

In our revision, we have
- Clarified that **we are only concerned with safety** and do not have a separate task objective in this work
- Added new experiments on pixel observations and higher dimensional parameter spaces (Figure 11)
- Clarified that Safe RL is orthogonal to our problem and perform new experiments to show how our method can be added to existing Safe RL methods (Appendix D.4, Figure 23)

After these additions, the reviewer **raised no further concerns**.

&nbsp;

# Key Additional Experiments & Clarifications

During the rebuttal, we provided extensive supplementary results to substantiate our claims. We hope you take these into account:
- **Pixel observations and high dimensional parameter spaces ($\a,\c$)**: We add two new environments (Figure 11), where our method outperforms the best baseline methods with almost **10\%** higher coverage gain in safety.
- **Application to Safe RL ($\a$)**: We add a new experiment to show how our method can be applied to not just PPO but also existing Safe RL methods (PPO-Lagrangian, SauteRL) to improve their safety rates by **>20%**.
- **Theoretical Analysis on Saddle-point Finding ($\b$)**: We clarify that the convergence guarantees for our saddle-point finding algorithm are **valid** on convex-concave functions, even when the expectation is approximated, and perform a new experiment comparing our method to Gradient Descent Ascent showing convergence (Figure 4).
- **Robustness Analysis of Saddle-point Finding to Violated Convex-Concavity ($\a$)**: We perform a new experiment showing that our method converges in practice even without convex-concavity and exact best-response oracles.
- **Theoretical Analysis on Classification Accuracy ($\c$)**: We prove new theoretical results that provide guarantees on classification accuracy that hold even when the policy $\pi$ changes.

&nbsp;
---

We hope this summary helps clarify the status of our submission. We are confident that FGE represents a significant step forward for tackling the problem of feasibility that has been underlooked in the RL safety community.

---

### Meta-Review · Area_Chair_cK2p · 2025-12-31

**Summary:**

The paper develops a new approach to expand the set of safe parameters in which a policy is safe. The rebuttal provided numerous clarifications and added new experiments, thereby strengthening the empirical evaluation. Overall, the paper has a solid contribution but would benefit significantly from a clearer presentation.

**Reviewer Concerns:**

- `CPF8`  was concerned with the relationship of the proposed method to safe RL algorithms. The rebuttal argues that the proposed approach is orthogonal to the safe RL problem. As an AC, I partially agree with the statement; however, the related work could still make the contrast from the problem under consideration to safe exploration \[1, 2\] and safe curriculum learning \[3, 4, 5\] from the safe RL literature clearer.
- `6h75` pointed out serious issues regarding the exposition of the problem setting. The paper went through a considerable revision, which partially addressed the concerns of the reviewer. Nevertheless, the presentation of the problem could still be refined to ensure it is more accessible to the machine learning community.
- `kNnA`
	1. had some concerns regarding the disparity between the assumptions of the theoretical claims and the application, where the $\pi$ and $\rho$ change over time. The authors acknowledge this is a relevant theoretical issue and argue that in practice it does not pose a problem.
	2. had concerns regarding the limitations of the approach to low-dimensional and deterministic settings. The rebuttal provided new experiments in problems with more parameters, which partially address the concern of the reviewer. Problems with stochastic dynamics are left open.



1. Berkenkamp, F., Turchetta, M., Schoellig, A. P., and Krause, A. (2017). Safe model-based reinforcement learning with stability guarantees. *NIPS*, 908–918.
2. Turchetta, M., Berkenkamp, F., and Krause, A. (2016). Safe exploration in finite Markov decision processes with Gaussian processes. *NIPS*, 4312–4320.
3. Eysenbach, B., Gu, S., Ibarz, J., and Levine, S. (2018). Leave no trace: Learning to reset for safe and autonomous reinforcement learning. *ICLR*.
4. Turchetta, M., Kolobov, A., Shah, S., Krause, A., and Agarwal, A. (2020). Safe reinforcement learning via curriculum induction. *NeurIPS*.
5. Koprulu, C., Simão, T. D., Jansen, N., and Topcu, U. (2025). Safety-prioritizing curricula for constrained reinforcement learning. *ICLR*.

**Reviewer Scores:**

- `CPF8`: 4 -> 6
- `6h75`: 6 -> 8
- `kNnA`: 6 -> 6

---

### Decision · Program_Chairs · 2026-01-26

Accept (Poster)